# You Point, I Learn: Online Adaptation of Interactive Segmentation Models for Handling Distribution Shifts in Medical Imaging

**Wentian Xu, Ziyun Liang, Harry Anthony, Yasin Ibrahim,**
**Felix Cohen, Guang Yang, Konstantinos Kamnitsas**
Department of Engineering Science
University of Oxford

## Abstract

Interactive segmentation uses real-time user inputs, such as mouse clicks, to iteratively refine model predictions. Although not originally designed to address distribution shifts, this paradigm naturally lends itself to such challenges. In medical imaging, where distribution shifts are common, interactive methods can use user inputs to guide models towards improved predictions. Moreover, once a model is deployed, user corrections can be used to adapt the network parameters to the new data distribution, mitigating distribution shift. Based on these insights, we aim to develop a practical, effective method for improving the adaptive capabilities of interactive segmentation models to new data distributions in medical imaging. Firstly, we found that strengthening the model's responsiveness to clicks is important for the initial training process. Moreover, we show that by treating the post-interaction user-refined model output as pseudo-ground-truth, we can design a lean, practical online adaptation method that enables a model to learn effectively across sequential test images. The framework includes two components: (i) a Post-Interaction adaptation process, updating the model after the user has completed interactive refinement of an image, and (ii) a Mid-Interaction adaptation process, updating incrementally after each click. Both processes include a Click-Centered Gaussian loss that strengthens the model's reaction to clicks and enhances focus on user-guided, clinically relevant regions. Experiments on 5 fundus and 4 brain-MRI databases show that our approach consistently outperforms existing methods under diverse distribution shifts, including unseen imaging modalities and pathologies. Code: https://github.com/WenTXuL/OAIMS

## 1 Introduction

Medical image segmentation facilitates disease analysis, diagnosis, and treatment. Deep-learning methods have driven notable advances in automated medical image segmentation (Azad et al., 2024). However, a major challenge is that the training-data distribution often differs from the test-data distribution—for example, images may be acquired on different scanners—severely hindering model performance. Although models lack knowledge about unseen test data distributions, human users (such as clinicians) are often still able to segment images in the target distribution with reasonable accuracy. Hence, their knowledge can be leveraged to guide models. Can we design an AI framework that enables models to be guided by human users in an easy, immediate, and continuous manner, so that they can effectively adapt to distribution shifts? Although not originally developed for solving data distribution shift problems, a class of deep learning models known as interactive segmentation models is well suited to this challenge.

Interactive segmentation models allow users to provide prompts, such as clicks, scribbles, or bounding boxes, which inform the model's prediction. A common strategy is to encode user prompts as additional input channels in convolutional networks, as seen in models like DeepIGeoS (Wang et al., 2018) and Interactive FCNN (Sakinis et al., 2019). More recent approaches, such as SAM (Kirillov et al., 2023), MedSAM (Ma et al., 2024), and Med-SA (Wu et al., 2023b), instead employ Transformers to encode user prompts. Both approaches have demonstrated strong performance on natural

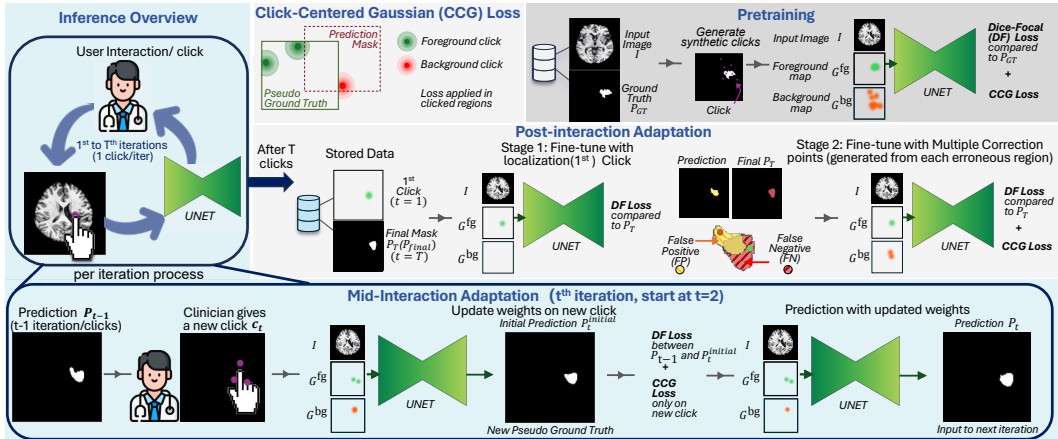

Figure 1: **Method overview.** For **Pretraining**, the model is trained with simulated clicks, provided as additional input channels besides the image. During **Inference** and adaptation, images arrive sequentially. For each image, the user iteratively provides $T$ clicks to correct the segmentation, until the final prediction $P_{\text{final}} = P_T$ is obtained. **Mid-interaction adaptation**: After each corrective click $c_t$, the model's output $P_t^{\text{initial}}$ is used as pseudo-label compared with the pre-correction output $P_{t-1}$ via the DF and CCG losses, to update model parameters. The updated model then produces refined output $P_t$, which is then shown to the user, ending iteration $t$. **Post-interaction adaptation:** Once the final corrected segmentation $P_T$ is obtained, it is used as pseudo-label to first fine-tune the model using a localization click (Stage 1), and then to fine-tune using multiple correction clicks, generated from areas where the prediction of Stage 1 disagrees with $P_T$ (Stage 2).

and medical images, highlighting the usefulness of incorporating user guidance. However, they do not include mechanisms for adapting model parameters from user corrections.

Our work explores how to leverage user guidance in interactive segmentation to improve performance under distribution shift most effectively. We identify that this requires complementary learning mechanisms for the pre-deployment training and post-deployment adaptation. For pre-deployment stage, we find that adding an optimization objective that enforces model predictions to align with user feedback in areas around given clicks improves performance under distribution shift.

We couple this with post-deployment learning mechanisms. **Post-deployment adaptation** methods for interactive segmentation **optimize a model for the specific data distribution encountered after deployment** using information from user prompts. They use **Online Learning** (Hoi et al., 2021) to update model parameters after each test image is processed sequentially [1]. Prior work on online adaptation for interactive medical image segmentation is limited. An early related method is IA+SA (Kontogianni et al., 2020), originally developed for natural images, which combines independent image-level adaptation (IA) and image-sequence adaptation (SA). Another recent related method, TSCA (Atanyan et al., 2024), achieves further improvements in performance after online adaptation. These methods leverage user corrections through sparse cross-entropy or focal loss, applied only to the clicked pixels. This focuses optimization narrowly on a small number of pixels while ignoring surrounding areas. Moreover, additional regularizers are often required to prevent overfitting to the few labeled pixels, increasing model complexity and the number of hyper-parameters.

Our work is based on the insight that, in a real-world interactive segmentation workflow where the user provides clicks to correct a model output, the segmentation predicted at the end of interactions should have sufficient quality to serve as pseudo ground-truth. We propose a Post-Interaction adaptation method based on using that final prediction as optimization target and show it results in effective adaptation without requiring complex regularizers. Furthermore, we generate *artificial* correction clicks from the pseudo-ground-truth mask and use them with the Click-Centered Gaussian (CCG) loss that we introduce to *strengthen the model's response around clicks* under the new data distribution. This improves performance in all tested distribution-shift scenarios.

---

[1] Adaptation in this context targets only the distribution seen post-deployment, thus does not need to handle Catastrophic Forgetting of past distributions as Continual Learning (Wang et al., 2024)

We further combine this with Mid-Interaction Adaptation, which adapts the model weights after each user click. We again rely on the segmentation mask predicted by the model and optimize the CCG loss, which emphasizes the region around the click. Unlike previous works that focus only on the clicked pixels for such adaptation, our method leverages the whole corrected segmentation mask together with the CCG loss, thereby optimizing over the greater surrounding region, improving adaptation performance.

We term the proposed method **OAIMS** (Online Adaptation for Interactive Medical-image Segmentation). Experiments under distribution shift across 5 fundus and 4 brain-MRI databases demonstrate that by using solely the proposed Post-Interaction method already results in adaptation performance that compares favorably to SOTA adaptation methods. When this is combined with Mid-Interaction adaptation, OAIMS consistently outperforms all previous methods, especially on brain MRI where Dice score improvements exceed 10%. Ablation studies show that the proposed CCG loss is consistently useful when employed in all 3 learning processes (pretraining, mid- and post- adaptation). Further analysis also shows strong robustness to settings that may cause overfitting to other methods.

## 2 METHODS

### 2.1 OVERVIEW: INTERACTIVE SEGMENTATION FRAMEWORK

We here provide an overview of the whole process, shown in Fig. 1, with the detailed algorithm of the process provided in the Appendix A.6. For simplicity, we describe it for binary segmentation, but it also applies for multiple classes, as shown in Experiments.

We define the interactive model as $f(I, C; \theta)$, where $I$ is the input image, $C$ is the set of user clicks, and $\theta$ are model parameters. A click is labeled either as foreground or background class. A foreground click indicates that the specific pixel belongs to the target object, background click indicates that it does not. We train the model on a source database with simulated clicks $C$. During inference, the model receives a sequence of images $\{I_1, I_2, \ldots, I_N\}$ from another database. For a single image $I_n$ separately, the user (or simulated user) first provides a localization click $c_1$ to trigger the interactive process. The *localization click* used to start interaction is simply a foreground click placed anywhere inside the target foreground object. The model predicts initial segmentation $P_1^n = f(I_n, c_1; \theta)$. Afterwards, multiple iterations of interactions occur. At iteration $t$ the user places a new click $c_t$ in a region where prediction $P_{t-1}^n$ is wrong. The click set is updated $C_t = C_{t-1} \cup \{c_t\}$, where $C_1 = \{c_1\}$. The model then predicts $P_t^n = f(I_n, C_t; \theta)$. Next interaction $t + 1$ then occurs, and so forth. After $T$ interactions we obtain the final prediction $P_T^n$, which we call $P_{\text{final}}^n$. While here $T$ is given a set value for simplicity, in a real-world setting $T$ would be as much as user requires to be satisfied with segmentation output. The whole process is then repeated for the next image $I_{n+1}$ in the sequence. For notational simplicity, we omit the image index $n$ in most formulas below.

During inference, we perform two types of online adaptation. The **Post-Interaction adaptation** is a two-stage method that updates the model after the iterative, interactive corrections for a single image have finished and the model has produced final segmentation $P_{\text{final}}$. This improves performance for subsequent images. **Mid-Interaction adaptation** happens after each interaction. It takes place before the $P_{\text{final}}$ is obtained. This strategy benefits both the current and subsequent images.

### 2.2 PRETRAINING THE INTERACTIVE MODEL

The base interactive model is a U-Net (Ronneberger et al., 2015) modified to accept both the image and click prompts as input. We use the same strategy as ICNN (Sakinis et al., 2019), where we set 2 guidance maps that encode foreground and background clicks, respectively, each with the same spatial dimensions as $I$. The raw guidance maps are zero everywhere except at clicked pixels; we then apply a Gaussian smoothing kernel and normalize each map to $[0, 1]$. These maps are concatenated with the image along the channel dimension. The concatenated tensor (image + foreground map + background map) is input to the model. We train the base model using simulated clicks and a compound loss: **Dice–Focal** (Eq. equation 4) and **CCG Loss** (Eq. equation 3). The CCG Loss proposed herein strengthens the model's response to user clicks. Combining Dice with Focal loss is beneficial in medical segmentation to handle the imbalanced number of background / foreground pixels. See Appendix A.2 for details regarding click simulation. We note that the backbone model,

here a Unet for its proven performance and computational efficiency Isensee et al. (2024), could be replaced with other architectures, since our online adaptation method is model-agnostic.

## 2.3 CLICK-CENTERED GAUSSIAN (CCG) LOSS

An interactive model should react to user clicks and update the surrounding region accordingly. We propose a Click-Centered Gaussian Loss to strengthen the model's reaction to clicks by penalizing wrong predictions near each click, weighted by a Gaussian kernel. This loss is employed in all three stages, pre-training, Post-Interaction adaptation, and Mid-Interaction adaptation.

Let $c$ denote a user click at pixel $(i', j')$ with class label $y_{i',j'} \in \{0, 1\}$. For any pixel $(i, j)$ we define the Gaussian weight and an indicator

$$G_c(i, j) = \begin{cases} \exp\left(-\frac{(i-i')^2 + (j-j')^2}{2\sigma^2}\right), & |i - i'| \leq 3\sigma \text{ and } |j - j'| \leq 3\sigma \\ 0, & \text{otherwise,} \end{cases} \tag{1}$$

$$I_c(i, j) = \begin{cases} 1, & P(i, j) = y_{i',j'}, \\ 0, & \text{otherwise.} \end{cases} \tag{2}$$

$P$ denotes the ground-truth mask (used for pretraining) or pseudo ground-truth mask (used for adaptation), and $P(i, j)$ is its pixel value at coordinates $(i, j)$.

Given the current prediction $\hat{P}$, the **CCG Loss** is

$$\mathcal{L}_{\text{CCG}} = \frac{\sum_{c \in C} \sum_{i=0}^{H-1} \sum_{j=0}^{W-1} G_c(i, j) \, I_c(i, j) \, \text{CE}\big(\hat{P}(i, j), \, P(i, j)\big)}{|C| HW} \tag{3}$$

where $C$ is the set of clicks for the current sample, $H \times W$ is the image size, and $\text{CE}(\cdot, \cdot)$ denotes cross-entropy loss. The penalty is applied only to pixels that *should* share the same class as the click using the indicator $I_c(i, j)$. For instance, given a foreground click, the loss only applies to surrounding pixels that are foreground in the ground truth mask.

**Why not apply the loss to all surrounding pixels?** Each click only serves to change the surrounding pixels to a specific target class. The click does not provide information for clusters of pixels belonging to a different class. Applying extra penalties to pixels annotated as another class in the ground truth may cause the model to overfit to specific regions or images, ultimately degrading overall performance when facing distribution shifts or performing online learning.

## 2.4 ONLINE ADAPTATION

We now describe the online adaptation process, which includes Post-Interaction and Mid-Interaction adaptation phases. All model parameters are updated in all phases.

**Post-Interaction Adaptation:** This process updates the model after a user has completed correcting the segmentation of an image $I$ using a set $C_T$ of $T$ clicks, resulting to final segmentation mask $P_{\text{final}} = P_T = f(I, C_T, \theta)$. The key assumption is that $P_{\text{final}}$ after the user finished their corrections is of "good enough" quality to serve as pseudo ground-truth mask for updating the model. In real-world practice this is rather easy to ensure, if we only adapt based on segmentations for which the user confirmed that the interactions led to satisfying output. Even if $P_{\text{final}}$ is imperfect, it still provides new information from users to update the model's knowledge.

The user interaction starts with a localization click. We therefore naturally split the post-interaction updates into: (i) Fine-tune with an initial localization click as input; (ii) Fine-tune with correction clicks as input.

**Stage 1 – Fine-tune with a Localization Click:** To enhance the model's ability to make a good initial segmentation on the new data, we first fine-tune it with one localization click $c_1$ as input, given by the user in the previous inference step. Given $c_1$, we obtain $P_1 = f(I, c_1, \theta)$. We then update the model by applying **Dice–Focal (DF) loss** (Milletari et al., 2016; Lin et al., 2017) to penalize deviations of $P_1$ from the user-corrected mask $P_{\text{final}}$, where

$$\mathcal{L}_{DF} = (1 - \alpha) \, \mathcal{L}_D + \alpha \, \mathcal{L}_F. \tag{4}$$

$\mathcal{L}_D$, $\mathcal{L}_F$ and $\alpha$ are the Dice loss, Focal loss, and a weighting hyper-parameter respectively. Only one Gradient Descent update is performed for each image using Eq. 4.

**Stage 2 – Fine-tune with Multiple Correction Clicks:** To improve the model's ability to leverage correction-clicks, we input a set of artificial correction-clicks $\hat{C}$, obtain model output $\hat{P} = f(I, \hat{C}, \theta)$ and update the model using $P_{\text{final}}$ as target. It is not appropriate to reuse the user's original correction clicks $C_T$, as they were already used to produce $P_{\text{final}} = f(I, C_T, \theta)$ –they would result in $\hat{P}$ being identical to $P_{\text{final}}$ and hence trivial updates. Instead, we generate a set $\hat{C}$ of clicks by comparing the Stage 1 output $P_1$ with $P_{\text{final}}$, locate false-positive and false-negative regions, and generate one artificial click in each erroneous connected component (up to $T$ clicks), without extra human input. These newly generated clicks $\hat{C}$ are fed to the model, yielding new prediction $\hat{P}$. We then apply the proposed CCG loss, along with Dice–Focal loss to penalize deviations of $\hat{P}$ from $P_{\text{final}}$ for further guidance. The total loss is:

$$\mathcal{L}_{\text{total}} = \mathcal{L}_{DF} + \beta \, \mathcal{L}_{CCG}. \tag{5}$$

The CCG loss ensures the model **reacts** to each click in its surrounding region during adaptation, which to our knowledge, no previous work addresses explicitly.

**Mid-Interaction Adaptation:** Besides Post-Interaction adaptation, we can also update the model *after each user click*. In this case, the updated parameters obtained after each click determine the next prediction, so the update not only improves segmentation of the following images but also of the current image. This in turn helps achieve a high quality $P_{\text{final}}$ after all $T$ interactions, which facilitates effective Post-Interaction adaptation afterwards, thus complementing and enhancing it.

We again leverage the idea of using as pseudo ground truth the model's output after correction by a user click. Let $P_{t-1} = f(I, C_{t-1}, \theta_{t-1})$ be the prediction after $t-1$ clicks. When the next corrective click $c_t$ is given, we obtain $P_t^{\text{initial}} = f(I, C_t, \theta_{t-1})$. We then optimize CCG plus Dice–Focal loss (Eq. 5) to penalize differences between $P_{t-1}$ and $P_t^{\text{initial}}$, using $P_{t-1}$ as the prediction and $P_t^{\text{initial}}$ as pseudo ground-truth. This formulation leverages the additional information induced by $c_t$ to the new state ($t$) in comparison to the previous state ($t-1$), to optimize model parameters. Here, after each user click, the loss $\mathcal{L}_{CCG}$ is applied to the latest click $c_t$. After the model parameters are updated to $\theta_t$, the model processes $C_t$ again and produces $P_t = f(I, C_t, \theta_t)$. $P_t$ is shown to the user (or simulator) to get the next click and serves as prediction for the next update. The process is repeated for each of the $T$ user clicks, until final prediction $P_{\text{final}} = P_T$ is obtained. Finally, Post-Interaction adaptation on that image is applied.

The pseudo ground truth $P_t^{\text{initial}}$ in Mid-Interaction adaptation is not perfect, so the CCG loss is very important. It helps the model to concentrate learning on regions close to the clicks, which are the most valuable and trustworthy areas. The CCG loss is not intended to strengthen the model's reaction here, because the $c_t$ is not used for obtaining $P_{t-1}$.

## 3 EXPERIMENTS

**Databases:** We evaluate our method on two types of data. **Fundus imaging:** We use 5 public databases: REFUGE2 (Orlando et al., 2020; Fang et al., 2022), G1020 (Bajwa et al., 2020), GS1-Drishti (Sivaswamy et al., 2014), GAMMA (Wu et al., 2023a), and PAPILA (Kovalyk et al., 2022). They are 2D RGB images acquired at different clinics with different scanners, hence each represents a different distribution. We perform multi-class segmentation {0: Background, 1: Outer-ring, 2: Cup}. We compute evaluation metrics on Cup and Disc, where Disc is the union of Outer-ring and Cup, as common in literature (Orlando et al., 2020). Unless stated otherwise, we treat $REFUGE2$ as the *source* database on which we pretrain the interactive model. We treat other databases as different *target* distributions for adaptation and evaluation. **MRIs of Brain Lesions:** We use 4 databases. Each contains a different type of pathology and some contain multiple MRI modalities: BRATS2023 - Glioma, with Flair, T1, T1c, T2 modalities (Baid et al., 2021); ATLAS v2.0 - Stroke, with T1 modality (Liew et al., 2022); WMH - white matter hyperintensities, with Flair and T1 (Kuijf et al., 2022); TBI - Traumatic Brain Injuries, with Flair and T1. TBI is the only non-public data we use. Each database is acquired from a different clinic, with different scanner. Each database and modality can therefore be regarded as a different data distribution. Although the MRI scans are 3D images, we test the models using 2D slices, by selecting the slice with the largest lesion area per case. For multi-class databases (BRATS, TBI), we merge all lesion classes into one label, and

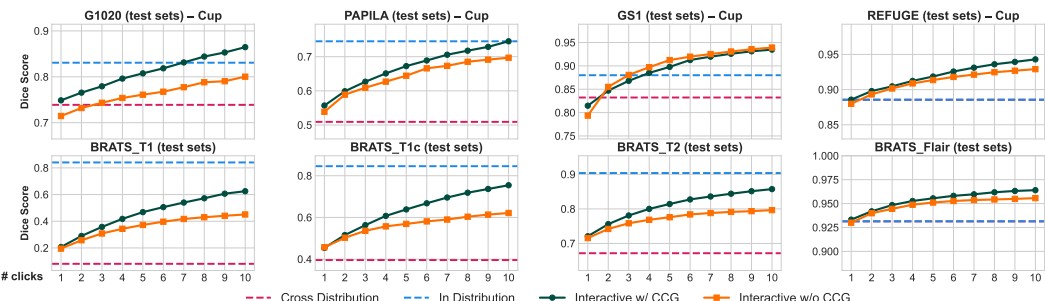

Figure 2: Dice-score performance for automatic and interactive models. All networks, except the in-distribution baseline, are trained on REFUGE (fundus) or BRATS-FLAIR (brain MRI). The x-axis represents the number of clicks. Horizontal lines mark automatic performance of automatic models in cross-distribution and in-distribution settings. Curves show the interactive segmentation model with and without the CCG loss; clicks significantly improve performance in all test cases, with the CCG loss providing additional gains, especially for large distribution gaps (e.g., BRATS-T1 ).

perform binary classification (healthy tissue VS lesion). We split BRATS into 1002 training cases and 249 test cases. Unless stated otherwise, we pre-train on the train split of BRATS using the Flair modality. We then use the other modalities of BRATS's test split, and all other databases, as the target distributions for adaptation and evaluation. All interactive models are trained with up to 10 simulated clicks per image, unless stated otherwise.

## 3.1 INTERACTIVE SEGMENTATION UNDER DISTRIBUTION SHIFT

We start with experiments that test the performance of the base interactive model under data-distribution shift, without any adaptation. We also assess if the proposed CCG loss helps an interactive segmenter to handle distribution shifts. We consider 2 types of distribution shifts. The first is across fundus databases. We consider REFUGE as the *source* distribution, and other fundus databases as *target* distributions. The second is across BRATS modalities. We consider BRATS FLAIR as the *source* distribution, and other BRATS modalities as *target* distributions.

Four models are compared for each of the 2 settings: (1) Automatic (non-interactive) U-Net (same backbone as the interactive model) trained on the source database, and applied to each of the target databases (cross-distribution); (2) Automatic model trained on the target database and applied to the target database (in-distribution). This is to quantify performance on target, without influence of distribution shift; (3) Interactive segmentation model trained on the source database without CCG loss, applied to each target database; (4) Similar to (3) but pretraining also uses CCG loss. Results are shown in Fig.2. We see a large difference between *in-distribution* and *cross-distribution* performance of automatic methods, across all settings, due to distribution shifts between *source* and *target* data. Nonetheless, ten clicks with the interactive segmentation model largely close the gap, confirming that interactive segmentation remains effective under strong shifts. Adding the CCG loss during pretraining yields improvements in most settings. This is because the CCG loss enforces the model to depend more on user input when given, signal unrelated to distribution shift, and less on the image signal where the shift manifests.

## 3.2 ONLINE ADAPTATION

We then evaluate our online adaptation methods, including *Mid-Interaction* and *Post-Interaction* adaptation. For all following experiments, during online adaptation, a sequence of images is input to the model. Each image is corrected through $T$ iterations (one click per iteration, $T = 10$ by default). For every image, the Dice score is calculated with the prediction mask at each iteration $t$ *after mid-interaction adaptation is performed in that iteration*. We measure average Dice score achieved for each image in the image sequence after 1, 5, and 10 interactions. The adaptation methods update the model after getting the segmentation result after each click and each image.

Table 1: Performance on fundus imaging (average Dice, %). Each image receives 10 clicks; the base interactive model is trained on REFUGE. Each cell reports Disc/Cup Dice at 1, 5, and 10 clicks. While ICNN* and Med-SA are non-adapting models, the other methods perform online adaptation. PI is our Post-Interaction adaptation strategy. PI+MI combines the Post-Interaction with the Mid-Interaction strategy.

| No. Cl | G1020 | | | PAPILA | | | GS1 | | | GAMMA | | |
|---|---|---|---|---|---|---|---|---|---|---|---|---|
| | 1 | 5 | 10 | 1 | 5 | 10 | 1 | 5 | 10 | 1 | 5 | 10 |
| ICNN* | 89.4/77.4 | 93.1/83.1 | 95.1/88.0 | 88.3/60.3 | 92.6/70.2 | 94.6/77.1 | 96.6/82.4 | 97.2/90.4 | 97.7/94.1 | 94.4/83.3 | 96.3/89.8 | 97.2/93.3 |
| Med-SA | 72.2/56.1 | 75.3/58.6 | 75.6/58.6 | 52.5/34.5 | 56.1/36.7 | 56.7/36.7 | 88.6/61.5 | 89.6/67.0 | 89.6/67.6 | 89.8/76.2 | 90.0/77.5 | 90.0/77.5 |
| IA+SA | 90.5/77.8 | 94.3/84.5 | 96.1/90.3 | 89.4/61.3 | 93.3/70.5 | 95.3/77.6 | 96.8/83.4 | 97.4/91.5 | 98.0/94.8 | 94.7/84.0 | 96.3/90.2 | 97.4/93.9 |
| TSCA | 89.9/77.6 | 94.2/85.0 | 96.1/90.7 | 89.6/61.7 | 94.0/72.2 | 96.1/79.0 | 96.9/85.4 | 97.5/92.9 | 98.0/95.5 | 94.6/84.2 | 96.4/90.7 | 97.5/**94.1** |
| PI | 93.5/81.9 | 95.9/88.5 | 97.0/92.2 | 94.1/73.0 | 96.1/80.0 | 97.0/85.8 | **97.3/89.5** | 97.7/94.1 | 98.2/95.7 | **95.3/85.9** | 96.5/**91.4** | 97.5/**94.1** |
| PI+MI | **93.6/82.2** | **96.4/90.0** | **97.5/92.7** | **94.7/73.0** | **96.5/81.0** | **97.5/86.2** | **97.3**/89.3 | **97.8/94.5** | **98.4/96.5** | 95.1/85.8 | **96.7**/91.3 | **97.9**/93.9 |

Table 2: Performance (Dice%) on different MRI modalities.

| No. Clicks | BRATS T1 | | | BRATS T1C | | | BRATS T2 | | |
|---|---|---|---|---|---|---|---|---|---|
| | 1 | 5 | 10 | 1 | 5 | 10 | 1 | 5 | 10 |
| ICNN* | 20.7 | 46.8 | 62.5 | 45.4 | 63.8 | 75.4 | 72.1 | 81.4 | 85.7 |
| Med-SA | 23.4 | 33.4 | 35.7 | 38.7 | 47.0 | 49.2 | 75.7 | 78.8 | 79.2 |
| IA+SA | 28.6 | 57.0 | 70.6 | 48.3 | 67.1 | 78.2 | 76.5 | 84.1 | 87.9 |
| TSCA | 34.4 | 60.9 | 74.0 | 50.1 | 70.2 | 79.6 | 77.7 | 85.8 | 89.0 |
| PI | 61.1 | 72.4 | 78.9 | 64.3 | 74.7 | 80.4 | 82.5 | 88.1 | 90.9 |
| PI+MI | **71.2** | **83.4** | **88.0** | **70.4** | **82.9** | **87.5** | **84.9** | **90.6** | **93.0** |

We set $\alpha = 0.7$ and $\beta = 200$, with $\sigma = 3$ in CCG loss, found adequate in preliminary experiments. The effect of different hyperparameter settings can be seen in Appendix A.3. We implement a base interactive model we denote as ICNN*, using a U-NET with the interactive method proposed by ICNN (Sakinis et al., 2019), and trained with our CCG loss. We implement IA+SA (Kontogianni et al., 2020) and TSCA (Atanyan et al., 2024) using the same pretrained base interactive model (with CCG loss) as our method for fair comparison. For all online adaptation methods, all model parameters are updated during adaptation. In addition, we include a SAM-based interactive medical image segmentation model, the Medical SAM Adapter (Med-SA) (Wu et al., 2023b), which is fine-tuned on the source data and frozen during testing (target data).

**Evaluation on Fundus data.** We pretrain the models on REFUGE as the source distribution, for multi-class segmentation. We then adapt and evaluate using each of the 4 other fundus databases separately as *target* distributions. Tab. 1 shows the average Dice for both disc and cup. On G1020 and PAPILA, where the data-distribution shift is large, our fast *Post-Interaction* method outperforms previous methods—especially on cups (disc segmentation is nearly perfect for most models and thus hard to improve)—and all adaptation approaches surpass the frozen base model. On GS1 and GAMMA, where the shift is small, our *Post-Interaction* method remains comparable or better. Using only the *Post-Interaction* adaptation, which requires two back-propagations, already surpasses previous methods that need more than ten back-propagations. Adding the Mid-Interaction adaptation gives slightly better results in most cases. The improvement becomes much more significant when facing large data-distribution shifts in the brain-MRI databases.

**Evaluation on MRI Modalities:** We here adapt our model to scenarios with larger distribution shifts – between different MRI modalities. The model is initially trained using the FLAIR scans of the training split. It is then adapted and evaluated on T1, T1c, and T2 scans of the test split (separate experiment per modality). As shown in Tab. 2, all online-adaptation methods outperform the base interactive model, ICNN*. Largest improvements shown in T1. Among online adaptation methods, our approach surpasses TSCA and IA+SA even with only Post-Interaction adaptation, especially when few clicks are given. Including Mid-Interaction adaptation yields even greater gains.

**Adapting to Different Brain Pathologies:** In addition, we test our model across different brain pathologies. We pretrain 2 models, one on BraTS Flair, and one on a combination of Flair/T1/T1c. We then adapt and evaluate the first on TBI-Flair and WMH-Flair, and the second on TBI-T1 and ATLAS-T1. Results are shown in Tab. 3. Even in these challenging settings, online-adaptation methods significantly boost performance, with our approach outperforming previous methods on all tasks. For TBI and WMH on FLAIR, our Post-Interaction method achieves results comparable to TSCA after 10 clicks but attains higher dice scores with fewer clicks. After adding Mid-Interaction adaptation, our method achieves significantly better results.For TBI-T1 and ATLAS, Post-Interaction alone

Table 3: Performance (Dice%) on different brain pathologies.

| | Trained on BRATS (Flair) | | | | | | Trained on BRATS (Flair, T1, T1c) | | | | | |
| | TBI Flair | | | WMH Flair | | | TBI T1 | | | ATLAS T1 | | |
| No. Clicks | 1 | 5 | 10 | 1 | 5 | 10 | 1 | 5 | 10 | 1 | 5 | 10 |
|---|---|---|---|---|---|---|---|---|---|---|---|---|
| ICNN* | 49.9 | 64.1 | 69.6 | 47.9 | 61.2 | 67.6 | 42.0 | 49.3 | 55.3 | 40.6 | 46.8 | 52.1 |
| Med-SA | 43.5 | 47.9 | 48.5 | 52.6 | 60.0 | 61.5 | 34.0 | 41.3 | 43.1 | 35.7 | 42.4 | 43.9 |
| IA+SA | 50.6 | 66.4 | 73.9 | 49.4 | 64.2 | 72.0 | 44.5 | 52.7 | 59.8 | 43.4 | 53.9 | 62.6 |
| TSCA | 52.7 | 66.1 | 73.7 | 52.8 | 66.7 | 72.7 | 44.4 | 55.8 | 63.9 | 43.4 | 55.7 | 64.0 |
| PI | 53.8 | 68.8 | 73.6 | 53.7 | 66.7 | 72.3 | **47.7** | 61.1 | 68.0 | 62.7 | 77.0 | 81.8 |
| PI+MI | **55.2** | **69.9** | **76.3** | **59.0** | **73.0** | **78.9** | **47.7** | **67.0** | **74.8** | **66.4** | **82.2** | **86.0** |

Table 4: Adapting with maximum 5 or 3 clicks per image.

| | 5 clicks | | | 3 clicks | | | |
| | BRATS | | | TBI | WMH | TBI | ATLAS |
| Method | T1 | T1c | T2 | Flair | Flair | T1 | T1 |
|---|---|---|---|---|---|---|---|
| ICNN* | 46.8 | 63.8 | 81.4 | 58.9 | 55.6 | 45.8 | 44.6 |
| TSCA | 62.9 | 69.0 | 86.0 | 61.9 | 59.5 | 52.9 | 51.4 |
| PI | 71.2 | 72.8 | 87.4 | 62.5 | 61.2 | 52.6 | 68.4 |
| PI+MI | **80.4** | **80.5** | **90.3** | **65.3** | **64.3** | **54.6** | **73.3** |

significantly outperforms previous methods, and Mid-Interaction further improves performance. Although the pseudo ground truth in early iterations is suboptimal, as shown in the table (low Dice score for 1 click), PI+MI can still learn from it and achieve higher scores.

We also observe that in all three tables, TSCA performs better than IA+SA, consistent with the previous studies (Atanyan et al., 2024). Thus, we compare only with TSCA in subsequent experiments for simplicity. Furthermore, we observe that in most cases, Med-SA performs significantly worse than the base interactive model, ICNN, across all three tables, especially after 10 points. Therefore, we do not employ the computationally expensive SAM-based model further.

**Adapting with fewer allowed corrections:** All previous experiments used $T = 10$ maximum clicks for correction of each image. However, a method should ideally also perform well with fewer maximum performed corrective interactions. Here, we test our online-adaptation methods on brain MRI using maximum $T = 3$ or $5$ clicks for interactive correction of each image. This also assesses the capability of our method to adapt using a less optimal pseudo ground-truth. Tab. 4 shows results using 5 or 3 clicks max per image, under the same experiment settings as Tab. 2 and Tab. 3. Even with fewer clicks, online-adaptation methods perform significantly better than the frozen model ICNN*. Our Post-Interaction (PI) adaptation continues to outperform previous methods in most cases. The addition of Mid-Interaction (PI+MI) further improves results. Although the model output after 3/5 clicks may be suboptimal, learning from it as pseudo ground-truth remains effective.

**Computational latency:** We evaluated the computational latency of our method to assess its practicality. On an NVIDIA A5000 GPU, updates are negligible (0.05s for MI; 0.09s for PI). Crucially, even on a CPU, latencies remain imperceptible (0.25s for MI; 0.41s for PI). This ensures a smooth workflow, confirming that our simple, effective design is well-suited for real-world application.

Table 5: Ablation study by including different terms in PI and MI, after 1, 5, 10 clicks. $CCGL_{MI}$ and $DFL_{MI}$ represent the Mid-Interaction adaptation. $CCGL_{PI}$ and $DFL_{PI}$ represent the stage 2 of the Post-Interaction adaptation. $S1_{PI}$ represent the stage 1 of the Post-Interaction adaptation.

| Loss terms | | | | | G1020 (Cup) | | | ATLAS | | | BRATS T1 | | | BRATS T2 | | |
| $DFL_{MI}$ | $CCGL_{MI}$ | $DFL_{PI}$ | $CCGL_{PI}$ | $S1_{PI}$ | 1 | 5 | 10 | 1 | 5 | 10 | 1 | 5 | 10 | 1 | 5 | 10 |
|---|---|---|---|---|---|---|---|---|---|---|---|---|---|---|---|---|
| ✓ | ✓ | ✓ | ✓ | ✓ | 82.2 | 90.0 | 92.7 | 66.4 | 82.2 | 86.0 | 71.2 | 83.4 | 88.0 | 84.9 | 90.6 | 93.0 |
| − | ✓ | ✓ | ✓ | ✓ | 82.1 | 88.9 | 93.0 | 65.9 | 82.0 | 85.8 | 69.4 | 81.9 | 86.8 | 84.3 | 90.4 | 92.8 |
| ✓ | − | ✓ | ✓ | ✓ | 81.5 | 87.0 | 90.0 | 65.2 | 80.1 | 83.8 | 42.3 | 46.6 | 48.9 | 84.6 | 90.4 | 92.6 |
| − | − | ✓ | ✓ | ✓ | 81.9 | 88.5 | 92.2 | 62.7 | 77.0 | 81.8 | 61.1 | 72.4 | 78.9 | 82.5 | 88.1 | 90.9 |
| − | − | ✓ | − | ✓ | 82.2 | 87.6 | 90.6 | 58.4 | 66.2 | 69.2 | 60.6 | 71.7 | 76.8 | 82.6 | 88.1 | 90.5 |
| − | − | − | ✓ | ✓ | 81.7 | 86.8 | 89.6 | 60.7 | 74.7 | 79.8 | 55.7 | 66.3 | 72.1 | 81.7 | 87.3 | 89.9 |
| − | − | − | − | ✓ | 81.6 | 87.7 | 91.4 | 59.0 | 73.0 | 78.7 | 48.6 | 61.3 | 67.9 | 80.9 | 86.4 | 89.4 |

**Ablation Study:** To evaluate the benefit of each component of our method, we conduct an ablation study on each term of our online adaptation method. The results are shown in Tab. 5. Dice scores are reported on four target databases: G1020 (cup), ATLAS, BraTS-T1, and BraTS-T2. The source databases are as follows: REFUGE2 for G1020 (cup), a combination of BraTS Flair/T1/T1c for

ATLAS, and BraTS Flair for both BraTS-T1 and BraTS-T2. The source-target pairs are consistent with previous experiments. The ablation terms are divided into two groups: PI (Post-Interaction adaptation) and MI (Mid-Interaction adaptation). **S1$_{PI}$** is the first stage of the Post-Interaction adaptation approach. **DFL$_{PI}$** and **CCGL$_{PI}$** are the second stage of the Post-Interaction adaptation processes with the Dice-Focal loss or Click-Centered Gaussian loss. **DFL$_{MI}$** and **CCGL$_{MI}$** are the Mid-Interaction adaptation processes with the Dice-Focal loss or Click-Centered Gaussian loss. Overall, the ablation study confirms the contribution of each component and stage in both the Mid-Interaction and Post-Interaction approaches. The two loss terms should be used together in each process. For example, on the challenging BraTS T1 dataset where tumor boundaries are not well defined, performing MI solely with Dice-Focal loss can cause the model to learn incorrect information, as segmentation errors often extend beyond cancer boundaries. Adding the CCG loss effectively mitigates this issue by focusing learning on the corrected areas.

We have seen that MI adds benefits on top of PI. But does the opposite also hold? We evaluate whether adapting with PI offers benefits when MI is already performed. With a budget of five clicks per image, Post-Interaction adaptation improves performance in nearly every scenario as show in Tab. 6. When the budget rises to ten clicks, Post-Interaction adaptation continues to provide substantial gains in the early iterations, but by the final click, its advantage narrows: WMH still benefits, while others do not. Exact numbers are given in the Appendix A.4. With more clicks, the model leans more heavily on MI, which may partially cover the updates supplied by Post-Interaction adaptation. Even though the influence of PI may diminish with extensive interaction provided, it helps users reach satisfactory results with *fewer* clicks, which is important for interactive workflows. We therefore recommend deploying both mechanisms in most situations.

Finally, we investigate aspects of the CCG loss in Tab. 7. Column "all" is the proposed version, "no_class" removes the class-limited mechanism of CCG (applies it to all surrounding pixels), and "no_gaussian" replaces the Gaussian kernel with a uniform kernel. Removing the Gaussian kernel or the class-limited mechanism reduces performance for both PI and PI+MI in most cases. Additionally, we investigated different values of $\sigma$ for the loss. As $\sigma$ approaches zero, the loss effectively reduces to a single-point focus. Ablation study that explores the effect of this parameter is in Appendix A.7. Results demonstrate the value of using a Gaussian over focusing on a single point.

Table 6: Ablation study for PI under a 5-click budget. (Average Dice% over 3 runs with different seeds.)

|  | BRATS | | | WMH | TBI |
|---|---|---|---|---|---|
| PI+MI | **80.4** | **80.5** | **90.3** | **70.7** | 68.9 |
| MI | 78.4 | 79.8 | 89.8 | 68.0 | **69.0** |
| No MI/PI | 46.8 | 63.8 | 81.4 | 61.2 | 64.1 |

Table 7: Ablation study on the design of CCG loss. Performance shown as Dice%.

|  | all | | no_class | | no_gaussian | |
|---|---|---|---|---|---|---|
|  | PI+MI | PI | PI+MI | PI | PI+MI | PI |
| BRATS (T2) | **93.0** | **90.9** | 92.1 | 87.7 | 90.6 | 90.1 |
| WMH | **78.9** | 72.3 | 77.6 | **73.1** | 72.2 | 69.8 |
| ATLAS | **86.0** | **81.8** | 80.7 | 75.7 | 79.8 | 80.5 |

### 3.3 ROBUSTNESS AND OVERFITTING

In this section, we conduct additional experiments to assess the robustness of our method and potential overfitting. When plenty of clicks are provided for an image, the model may overfit to that image. To explore this, we consider an extreme case where each image receives 50 clicks (TSCA (50), OAIMS (50)). The result is shown in Tab. 8. In this scenario, TSCA (50) exhibits lower performance at the early clicks (e.g., click 1, 3) compared to TSCA(10), indicating potential overfitting to previously seen images. In contrast, our method performs significantly better with 50 clicks compared to 10, and does not exhibit signs of overfitting.

We also examine a challenging scenario, where images from different databases—BraTS T1, BraTS T2, and WMH FLAIR—are randomly shuffled together, with each database contributing 25 images. The result is shown in Tab. 9. Despite substantial domain differences, our method continues to outperform both the ICNN* and TSCA. Notably, while TSCA's performance approaches that of ICNN*, our method maintains a clear advantage. Removing Post-Interaction adaptation leads to performance drops. Hence our adaptation approach allows clinicians to use a single model that adapts to multiple diseases simultaneously, eliminating the need to manage multiple models.

Because our method uses model predictions as pseudo ground-truth, it is natural to wonder if the process would accumulate errors and corrupt the model over time in scenarios when predictions are

Table 8: Performance on BRATS T1 with a budget of 10 or 50 clicks per image. Dice shown at 1, 3, 10, 20, and 50 clicks. Overfitting past images lowers Dice on next image with few clicks (1–3) using TSCA but not our method.

| No. Clicks | 1 | 3 | 10 | 20 | 50 |
|---|---|---|---|---|---|
| ICNN* | 22.1 | 35.8 | 62.9 | 78.7 | 87.1 |
| TSCA(50) | 28.9 | 49.6 | 75.3 | 88.4 | 92.9 |
| TSCA(10) | 34.4 | 51.4 | 74.0 | N/A | N/A |
| OAIMS (50) | 73.9 | 81.8 | 89.8 | 94.3 | 95.9 |
| OAIMS (10) | 61.1 | 68.9 | 78.9 | N/A | N/A |

Table 9: Adapting to a database composed of images from BRATS T1, T2, and WMH FLAIR.

| No. Clicks | 1 | 5 | 10 |
|---|---|---|---|
| ICNN | 50.1 | 65.6 | 74.3 |
| TSCA | 52.0 | 67.1 | 75.3 |
| OAIMS(MI) | 52.1 | 72.1 | 80.6 |
| OAIMS(PI+MI) | 55.5 | 73.7 | 81.6 |

bad or the user provides wrong clicks by mistake. We conducted 3 experiments to assess this. First, to evaluate how the method performs when model predictions are of low quality (hence erroneous pseudo ground-truth), we test a scenario with **extreme domain shift**. Specifically, we pretrained the model using only BRATS Flair and apply it on Atlas T1. Results are shown in Table 10 (top). This scenario represents a very large domain gap, where the base interactive model (ICNN*) without online adaptation, achieves very poor initial segmentation: with just 1 localisation click, it achieves only 9.3% Dice. After 3 and 10 clicks, the ICNN* still performs very poorly (12% and 24.3% Dice respectively), although it does improve slightly with each click. We then apply online adaptation to the above model in 3 settings, when only 1, 3, or 10 clicks are allowed. Our method improves performance in all settings, and recovers high 80% Dice when using 10 clicks. To further assess robustness, we make this scenario even more challenging by ordering the images such that **hardest images** (lowest segmentation Dice) **are presented first in the sequence**. Here, the initial pseudo ground truth of the early images is very poor, with Dice near 0%. Results in Table 10 (bottom) show that performance of our method is comparable to random ordering, indicating that extreme initial failures do not destabilize the model for subsequent images. Finally, we assess robustness to **noisy input**, such as in the case of wrong user clicks. We simulated erroneous clicks by generating clicks in areas that the model assigned correct class, but we assign the opposite class to the click (i.e. ask for wrong class correction). Results in Table 11 show that our online adaptation method improves over a non-adaptive ICNN*, achieving good performance on BRATS T1 even when 40% of clicks are given the wrong class, when first 4 correction clicks per image are given wrong, or when for 40% of images all clicks are given wrong, demonstrating resilience against user errors.

Table 10: Robustness to pseudo ground-truth of bad quality due to extreme domain shift (top) and "worst case first" ordering (bottom).

| No. Clicks | 1 click | 3 clicks | 10 clicks |
|---|---|---|---|
| ICNN* (No Adapt) | 9.3 | 12.8 | 24.3 |
| TSCA | 9.3 | 40.7 | 65.6 |
| OAIMS (Ours) | **25.6** | **57.9** | **80.0** |

| No. Clicks | 1 | 5 | 10 |
|---|---|---|---|
| OAIMS (Random Order) | 67.5 | 82.2 | 86.0 |
| OAIMS (Worst first) | 66.4 | 81.6 | 85.3 |

Table 11: Robustness to noisy (wrong) input clicks.

| Noise Type | | 1 | 5 | 10 |
|---|---|---|---|---|
| 40% clicks are wrong | ICNN* | 20.6 | 32.9 | 43.9 |
| | OAIMS | **67.8** | **71.8** | **74.6** |
| First 4 clicks wrong on each image | ICNN* | N/A | 13.3 | 43.3 |
| | OAIMS | N/A | **56.9** | **75.1** |
| 40% images get only wrong clicks | ICNN* | 21.0 | 40.4 | 53.8 |
| | OAIMS | **68.1** | **75.2** | **76.9** |

## 4 CONCLUSION

This study investigates how to train and adapt an interactive segmentation model for medical imaging to better handle data distribution shifts. We proposed an online adaptation framework that integrates both Post-Interaction and Mid-Interaction approaches, enabling the model to continuously adapt to new data distributions. A Click-Centered Gaussian loss is proposed, which enhances the model's responsiveness to user inputs. We demonstrate the effectiveness of our method through extensive experiments with diverse distribution shifts. The promising performance underscores the transformative potential of adaptive interactive segmentation in advancing both clinical practice and research applications. The methodology is amenable to other types of user inputs beyond clicks, such as scribbles, and different types of backbone models, which could be explored in future work.

## ACKNOWLEDGMENTS

ZL is supported by scholarship provided by the EPSRC Doctoral Training Partnerships programme [EP/W524311/1]. HA is supported by a scholarship via the EPSRC Doctoral Training Partnerships programme [EP/W524311/1, EP/T517811/1]. YI is supported by the EPSRC Centre for Doctoral Training in Health Data Science (EP/S02428X/1). The authors would also like to thank Dr. Yu Liu for his valuable help on this paper.

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

# A APPENDIX

## A.1 VISUALIZATION RESULTS

Fig. 3 presents the visualization results demonstrating the adaptation performance on the BRATS dataset. The segmentation map is overlaid in red on the original image. Our OAIMS (PI+MI)

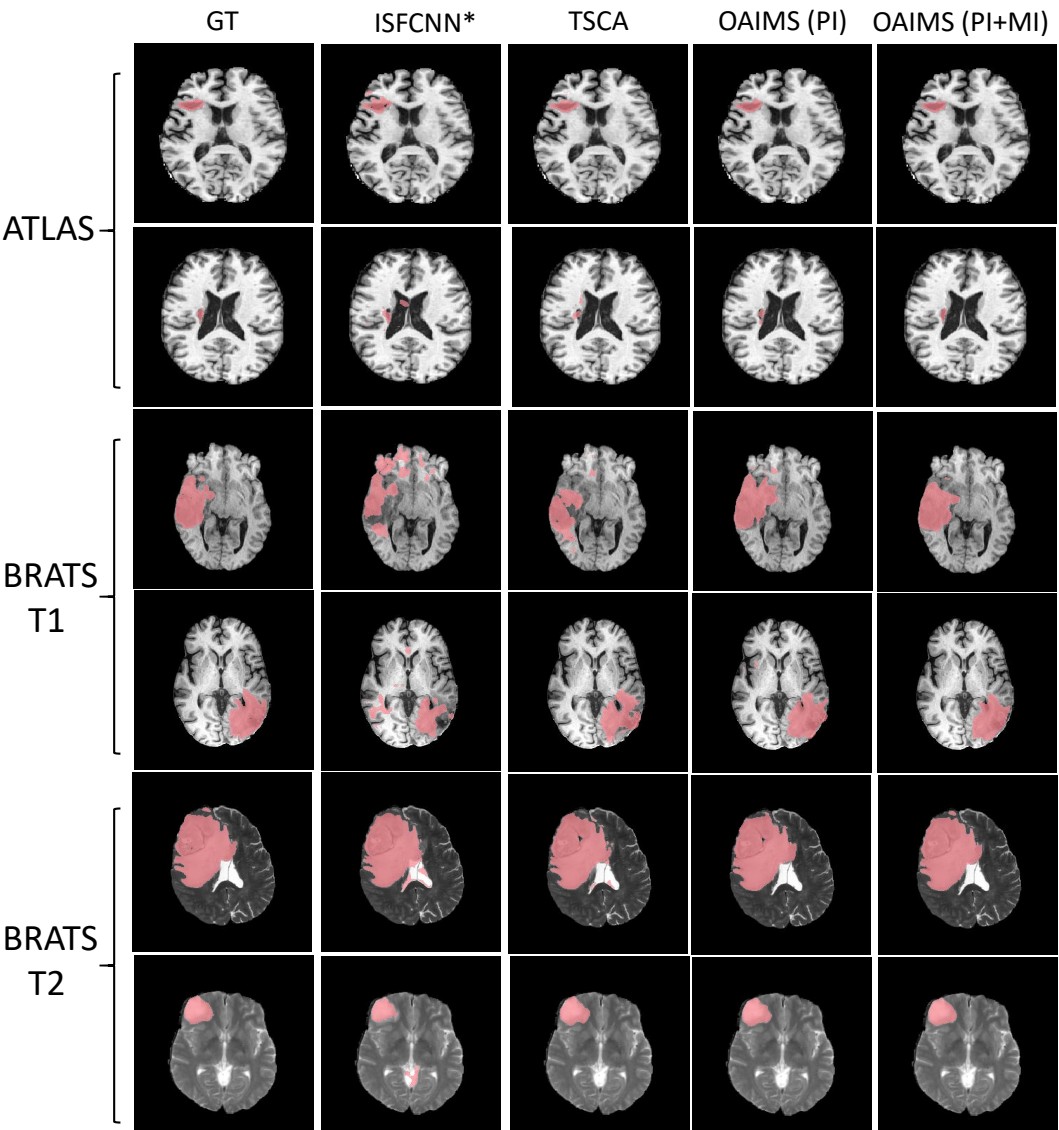

Figure 3: Visualizations on BRATS Databases demonstrating adaptation to different modalities and pathologies (Trained on BRATS Flair for BRATS T1 and BRATS T2, Trained on BRATS Flair/T1/T1c for ATLAS

method produces segmentations that are closest to the ground truth (GT), with more accurate boundaries compared to other methods.

Fig. 4 illustrates the databases used in our experiments. This visualization helps to better understand the distribution shifts across different databases and modalities.

Fig. 5 illustrates how the predicted segmentation evolves across interaction clicks.

## A.2 DETAILS OF THE SIMULATION PROCESS

To train and evaluate the interactive model, we simulate user interactions with an automatic point-generation procedure that places clicks in incorrectly segmented regions.

During training, we first generate a random click inside the target foreground object as a localization click. Based on the resulting segmentation mask and the ground truth, we identify incorrectly

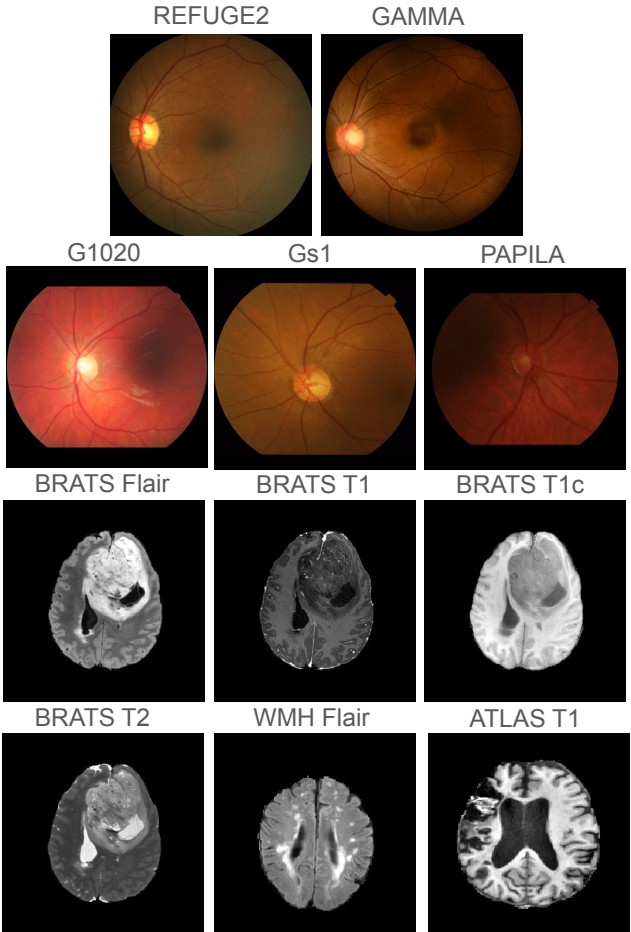

Figure 4: Illustration of the databases used. The distribution across different datasets and modalities can be visually observed.

segmented regions with connected components. Each erroneous component is ranked by size, and a random point is generated within each. We then select the first $K$ points from this queue, where $K$ is the desired number of clicks, and feed them into the model simultaneously. In our training, $K$ is randomly sampled for each iteration from a uniform distribution in the range $[1, 10]$.

At inference time, user clicks are simulated iteratively. First, one random click is placed inside the target foreground object. Then, based on the predicted segmentation and the ground truth, a correction click is placed in the largest erroneous component (including both false positives and false negatives). A new segmentation is generated using all previous clicks, and the process repeats until a total of $K$ clicks is reached.

For the simulation process in the Post-Interaction stage, the procedure is similar to training. However, instead of using the real ground truth, this step relies on a pseudo ground-truth mask.

### A.3 EFFECT OF DIFFERENT HYPERPARAMETER SETTINGS

In this section, we evaluate the effect of three hyperparameters, $\alpha$, $\beta$, and $\sigma$, on the performance of our method. The results are presented in Table 12.

We observe that relatively small values of $\beta$ (e.g., 100) and $\sigma$ (e.g., 1) lead to noticeable performance drops on the BRATS T1 dataset. This suggests that overly small values can hinder the model's ability to effectively utilize the CCG loss. Therefore, we recommend selecting values greater than 100 for $\beta$ and greater than 1 for $\sigma$. The choice of $\alpha$ also influences performance on BRATS T1, while its

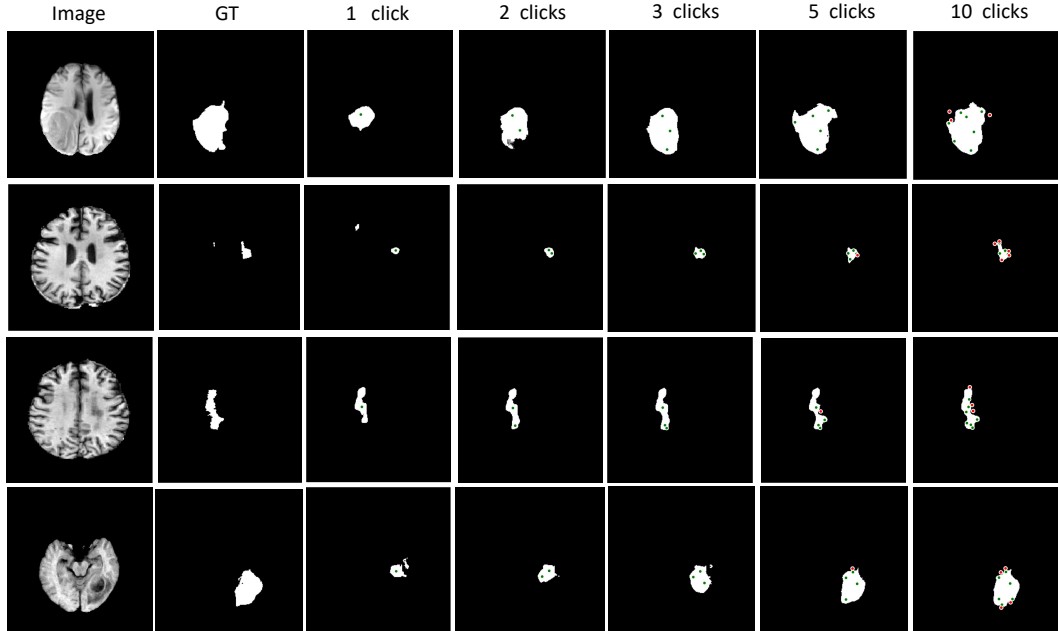

Figure 5: Illustration of how the predicted segmentation from our OAIMS (PI+MI) method evolves as more interaction clicks are provided. The examples are brain MRI images (ATLAS and BRATS T1). From left to right: input image, ground truth, and predictions with 1, 2, 3, 5, and 10 clicks. The results show progressive refinement of the segmentation as more clicks are provided.

impact on WMH is minimal. We do not specifically tune $\alpha$ to achieve the best results on BRATS T1.

Overall, performance remains stable in most cases. Note that in our experiments, when comparing with other methods, we did not perform hyperparameter tuning to obtain the best-performing configuration for any specific database. This decision was made due to the relatively stable performance of our method across tasks and to better demonstrate its robustness.

Table 12: Dice scores (%) on WMH and BRATS T1 under a 10-click interaction budget. Each value of $\alpha$, $\beta$, and $\sigma$ is tested in combination with all values of the other two hyperparameters (i.e., 3 $\alpha$ × 4 $\beta$ × 3 $\sigma$ total combinations). The reported score for a parameter's value is the average over all combinations that include it.

| Parameter | Value | BRATS T1 (%) | WMH (%) |
|---|---|---|---|
| $\alpha$ | 0.3 | 85.6 | 77.7 |
| | 0.5 | 87.0 | 77.6 |
| | 0.7 | 84.3 | 78.3 |
| $\beta$ | 100 | 81.3 | 77.5 |
| | 200 | 86.9 | 77.9 |
| | 300 | 87.2 | 78.1 |
| | 400 | 87.4 | 78.1 |
| $\sigma$ | 1 | 81.0 | 77.2 |
| | 3 | 87.9 | 78.2 |
| | 5 | 88.1 | 78.3 |

## A.4  ABLATION STUDY ON PI (10 CLICKS IN TOTAL)

As supplementary information to the main paper, we provide the numerical results of the ablation study of Post-Interaction (PI) adaptation under a budget of 10 clicks. The results are shown in Tab. 13. PI continues to offer substantial performance gains in the early iterations; however, by the final click, its advantage diminishes—WMH still benefits, while other methods do not. We continue to recommend deploying both mechanisms in most situations, with further explanation provided in the main paper.

Table 13: Ablation study for PI with a 10-click budget. Dice shown at 1, 3, 10 clicks. PI and MI are the proposed Post-Interaction process and the Mid-Interaction process of our method.

| Dataset | PI+MI | | | MI | | |
|---|---|---|---|---|---|---|
| | 1 | 3 | 10 | 1 | 3 | 10 |
| BRATS T1 | 71.2 | 79.4 | 88.0 | 69.1 | 77.7 | 87.6 |
| BRATS T1c | 70.4 | 78.9 | 87.5 | 68.3 | 77.9 | 87.7 |
| BRATS T2 | 84.9 | 88.6 | 93.0 | 84.6 | 88.6 | 93.0 |
| WMH Flair | 58.9 | 68.6 | 78.9 | 58.4 | 68.1 | 78.0 |
| TBI Flair | 55.2 | 65.6 | 76.3 | 53.3 | 64.4 | 76.4 |

## A.5  DICE SCORE

To evaluate segmentation performance, we use the Dice score. It measures the overlap between the predicted segmentation mask $P$ and the ground truth $G$, and is defined as:

$$\mathrm{Dice}(P, G) = \frac{2|P \cap G|}{|P| + |G|} = \frac{2TP}{2TP + FP + FN} \tag{6}$$

Here, $TP$, $FP$, and $FN$ denote the number of true positives, false positives, and false negatives, respectively. A higher Dice score indicates a greater overlap between the prediction and the ground truth.

The Dice score is particularly well-suited for medical image segmentation, as it emphasizes accurate delineation of foreground regions—such as lesions—which are often small and sparse. As a result, we adopt Dice as our evaluation metric.

## A.6  ALGORITHMIC DESCRIPTION

We here present the algorithmic details of the source training procedure (Algorithm 2) and the test-time inference/adaptation loop (Algorithm 1).

Although Algorithm 1 is formulated with a fixed number of interactions ($T$) for simplicity, consistent with the main paper, in clinical practice, the user can provide any number of clicks for each image until they are satisfied.

---

**Algorithm 1** Online Adaptation for Interactive Segmentation (OAIMS)

---

1: **Input:**
2: $f$: Base interactive Model
3: $\theta^*$: Pretrained parameters
4: $\mathcal{S}$: A sequence of images $I$
5: $T$: Number of user-interaction clicks

6: $\theta \leftarrow \theta^*$
7: **for** $I \in \mathcal{S}$ **do**                       ▷ Process each image sequentially
     — **Inference and Mid-Interaction (MI) Adaptation** —
8:     $t = 1$
9:     $c_1 \leftarrow \text{LocalizationClick}(I)$         ▷ Get localization click from target foreground object
10:     $C \leftarrow \{c_1\}$
11:     $P_1 \leftarrow f(I, C; \theta)$
12:     **for** $t = 2$ **to** $T$ **do**
13:        $c_t \leftarrow \text{Getclick}(P_{t-1}, I)$                  ▷ User provides new click
14:        $C \leftarrow C \cup \{c_t\}$
15:        $P_t^{\text{initial}} \leftarrow f(I, C; \theta)$    ▷ Get $P_t^{\text{initial}}$ for updating the model (no gradient calculation)
16:        $\mathcal{L}_{MI} \leftarrow \mathcal{L}_{DF}(P_{t-1}, P_t^{\text{initial}}) + \beta\mathcal{L}_{CCG}(P_{t-1}, P_t^{\text{initial}}, c_t)$
17:        $\theta \leftarrow \text{UpdateParameters}(\theta, \mathcal{L}_{MI})$
18:        $P_t \leftarrow f(I, C; \theta)$
19:     **end for**
20:     $P_{final} \leftarrow P_T$                           ▷ Final mask after T clicks
     — **Post-Interaction (PI) Adaptation stage1** —
21:     $C \leftarrow \{c_1\}$                             ▷ Get first click from MI
22:     $P_1 \leftarrow f(I, C; \theta)$
23:     $\mathcal{L}_{S1_{\text{PI}}} \leftarrow \mathcal{L}_{DF}(P_1, P_{final})$
24:     $\theta \leftarrow \text{UpdateParameters}(\theta, \mathcal{L}_{S1_{\text{PI}}})$
     — **Post-Interaction (PI) Adaptation stage2** —
25:     $\hat{C} \leftarrow \text{GenerateClicks}(P_1, P_{final}, T)$     ▷ Generate on each erroneous component (up to T)
26:     $\hat{P} \leftarrow f(I, \hat{C}; \theta)$
27:     $\mathcal{L}_{S2_{\text{PI}}} \leftarrow \mathcal{L}_{DF}(\hat{P}, P_{final}) + \beta\mathcal{L}_{CCG}(\hat{P}, P_{final}, \hat{C})$
28:     $\theta \leftarrow \text{UpdateParameters}(\theta, \mathcal{L}_{S2_{\text{PI}}})$
29: **end for**

---

**Algorithm 2** Pretraining the Interactive Model

---

1: **Input:**
2: $f$: Base interactive Model
3: $\mathcal{D}_{\text{source}}$: Source (training) dataset (e.g. REFUGE). $(I, M_{gt})$: Image and ground-truth mask
4: $T_{\max}$: Max number of simulated clicks (e.g. 10)

5: **for** $(I, M_{gt}) \in \mathcal{D}_{\text{source}}$ **do**
6:     $c_{loc} \leftarrow \text{SimulateLocalizationClick}(M_{gt})$
7:     $P_{loc} \leftarrow f(I, \{c_{loc}\}; \theta)$                   ▷ Get prediction from localization click
8:     $E \leftarrow \text{FindErroneousRegions}(P_{loc}, M_{gt})$        ▷ Find all erroneous connected components
9:     $C_{ranked} \leftarrow \text{RankAndGenerateClicks}(E)$
                     ▷ Rank components by size, randomly generate one click in each component
10:    $K \sim \mathcal{U}\{1, T_{\max}\}$           ▷ Sample $K$ from a discrete uniform distribution
11:    $C \leftarrow \text{SelectFirstM}(C_{ranked}, K)$             ▷ Select the top $K$ error clicks
12:    $\hat{P} \leftarrow f(I, C; \theta)$
13:    $\mathcal{L} \leftarrow \mathcal{L}_{DF}(\hat{P}, M_{gt}) + \mathcal{L}_{CCG}(\hat{P}, M_{gt}, C)$
14:    $\theta \leftarrow \text{UpdateParameters}(\theta, \mathcal{L})$
15: **end for**
16: **return** $\theta$

---

### A.7 Ablation Study of $\sigma$ in CCG Loss

We conducted an ablation study on the BRATS-T1 dataset by varying the standard deviation $\sigma$. We evaluated $\sigma \in \{0, 1, 2, 3, 4, 5\}$, where the $\sigma = 0$ case was implemented as applying the CCG loss only on the single pixel. The results are illustrated in Figure 6.

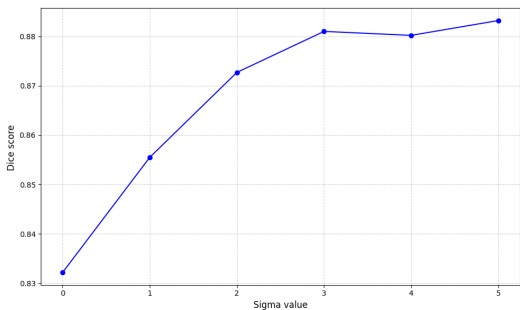

Figure 6: Ablation study of the Gaussian parameter $\sigma$ of CCG Loss on the BRATS-T1 dataset.

As $\sigma$ increases from 0 to 3, the performance improves significantly, as the model receives more information from surrounding pixels. When $\sigma$ is large enough (e.g., $\sigma \geq 3$) the method seems robust to the choice of its value, achieving stable performance.

We also observed that for $\sigma = 0$ the adaptation was highly unstable, sometimes resulting in very low performance. For example, over 3 runs (RNG seeds) of the same experiment, the final score was 79.9, 46.9, and 83.2. This instability was only observed for $\sigma = 0$. This is likely because the model can easily overfit a single pixel, which can strongly hinder performance.

### A.8 Assessing Number of Clicks to Reach Target Dice

To further evaluate the clinical efficiency of our method, we measure number of clicks to reach a specified Dice score, which assesses how quickly a user can achieve satisfactory segmentation.

In this experiment, we set a target Dice score of 80% with a maximum of 20 clicks per image. If an image failed to reach the target within 20 clicks, its click count was capped at 20. We compared our proposed method (PI+MI) against the strongest baseline, TSCA, on the BraTS dataset (transferring from Flair to T1, T1c, and T2).

The results are presented in Table 14. Our method requires significantly fewer clicks to reach the target accuracy across all modalities, demonstrating superior efficiency in correcting domain shifts.

Table 14: Average number of clicks required to reach 80% Dice (lower is better), with a maximum of 20 clicks per image

| Method | BRATS T1 | BRATS T1C | BRATS T2 |
|---|---|---|---|
| TSCA | 10.60 | 8.71 | 3.63 |
| PI+MI (Ours) | **4.43** | **4.59** | **2.33** |

### A.9 Comparison with Applying Unsupervised Losses on Model Prediction

Given that our method is based on the idea of using the user-refined model predictions as pseudo-labels, one may wonder if it would be sufficient to simply minimize Cross-Entropy (CE) between the model prediction and the pseudo-label, a method often used for learning from unlabeled data Lee et al. (2013). Specifically, assume for an input image $I$, after $t$ corrective clicks by the user, the model with parameters $\theta_t$ predicts $p_t \in \mathbb{R}^{H \times W \ K}$ the class-probability maps (output of softmax in our model) and $P_t = argmax(p_t) \in \mathbb{R}^{H \times W}$ the predicted segmentation. Here, H and W are the height and width of $I$ and these maps, and $K$ the number of classes in the segmentation task. Then, $p_t^k \in \mathbb{R}^{H \times W}$ is the class-probability map for class $k$, $p_t^k(i, j) \in [0, 1]$ the probability that the pixel with coordinates $(i, j)$ is of class $k$, and $P_t(i, j)$ the predicted class for the pixel. Then,

to optimize model parameters $\theta_t$, this method minimizes for each image the Cross-Entropy loss between predicted class-probabilities $p_t$ and the pseudo-label mask $P_t$:

$$\mathcal{L}_{\text{CE}} = -\sum_{i,j} \sum_k \mathbb{1}_{[k=P_t(i,j)]} \log p_t^k(i,j), \qquad (7)$$

where $\mathbb{1}_{[A]}$ is the indicator function that takes value 1 if $A$ is $True$, i.e. only for $k$ equal to the predicted label) and 0 otherwise.

Another very related method is Entropy-Minimization for learning from unlabeled data Grandvalet & Bengio (2004). Using similar notation as above, the Entropy loss for an image is defined as:

$$\mathcal{L}_{\text{Ent}} = -\sum_{i,j} \sum_k p_t^k(i,j) \log p_t^k(i,j), \qquad (8)$$

We experiment with these methods by replacing our own methodology during PI and MI stages instead of our methodology. For PI, we process one image at a time (as for OAIMS) and after $T$ user corrections per image, we obtain the final segmentation predicted by the model $P_{\text{final}} = P_T$ and associated probability maps $p_T$. We then apply an update to model parameters, similar as in our method, by minimizing one of the two above losses (using $P_T$ as pseudo-label in CE, whereas EM does not require it). We then experiment with using these losses both during MI and PI. In this case, after each user click $t$, we optimize one of the above losses, and after all $T$ corrections were completed, the losses are also optimized for PI as above.

We performed these experiments based on the Fundus and BRATS datasets. Hyper-parameters of these unsupervised methods were optimized via cross-validation to get optimal performance from them. As shown in Table 15, these losses provide almost no improvement and sometimes even yield worse performance than our non-adaptive backbone model (ICNN*). This is likely because the information gained from unsupervised losses is limited. Our method, OAIMS, outperforms consistently, especially with fewer clicks. When the distribution shift is larger, between BRATS datasets (Table 16), Online learning methods such as IA+SA and our OAIMS outperform unsupervised losses in all cases with much greater difference ( 5-60% Dice dependent on setting), especially on BRATS T1 and T1c. These experiments show that these unsupervised losses by themselves are not sufficient. Although they take advantage of the corrected model predictions, they do not explicitly use the user clicks. Instead, it is valuable to design frameworks that effectively leverage signal from user interactions to adapt to new data, like our OAIMS framework that explicitly emphasizes areas around user clicks via CCG.

Table 15: Performance on fundus imaging.

| No. Clicks | G1020 | | | PAPILA | | | GS1 | | | GAMMA | | |
|---|---|---|---|---|---|---|---|---|---|---|---|---|
| | 1 | 5 | 10 | 1 | 5 | 10 | 1 | 5 | 10 | 1 | 5 | 10 |
| ICNN* | 89.4/77.4 | 93.1/83.1 | 95.1/88.0 | 88.3/60.3 | 92.6/70.2 | 94.6/77.1 | 96.6/82.4 | 97.2/90.4 | 97.7/94.1 | 94.4/83.3 | 96.3/89.8 | 97.2/93.3 |
| Entropy-Min PI | 89.4/77.2 | 93.1/83.1 | 95.0/87.9 | 88.7/60.4 | 92.7/70.4 | 94.9/77.3 | 96.7/82.5 | 97.2/90.5 | 97.7/94.1 | 94.4/83.5 | 96.1/89.6 | 97.2/93.0 |
| Entropy-Min MI+PI | 89.7/77.7 | 93.3/83.0 | 95.1/87.4 | 89.6/61.2 | 93.3/70.4 | 95.1/76.8 | 96.6/82.1 | 97.1/90.0 | 97.6/93.7 | 94.5/83.8 | 96.1/89.7 | 97.2/93.0 |
| Cross-Entropy PI | 89.2/77.4 | 93.1/83.1 | 95.1/88.2 | 88.3/60.3 | 92.6/70.3 | 94.8/76.7 | 96.6/82.6 | 97.1/90.8 | 97.6/94.1 | 94.4/83.5 | 96.2/89.6 | 97.1/93.0 |
| Cross-Entropy MI+PI | 89.6/77.7 | 93.3/83.3 | 95.0/87.4 | 89.5/61.1 | 93.1/70.6 | 95.1/76.8 | 96.7/83.7 | 97.1/90.7 | 97.7/93.9 | 94.5/83.8 | 96.2/89.9 | 97.1/93.1 |
| IA+SA | 90.5/77.8 | 94.3/84.5 | 96.1/90.3 | 89.4/61.3 | 93.3/70.5 | 95.3/77.6 | 96.8/83.4 | 97.4/91.5 | 98.0/94.8 | 94.7/84.0 | 96.3/90.2 | 97.4/93.9 |
| TSCA | 89.9/77.6 | 94.2/85.0 | 96.1/90.7 | 89.6/61.7 | 94.0/72.2 | 96.1/79.0 | 96.9/85.4 | 97.5/92.9 | 98.0/95.5 | 94.6/84.2 | 96.4/90.7 | 97.5/**94.1** |
| OAIMS (PI+MI) | **93.6/82.2** | **96.4/90.0** | **97.5/92.7** | **94.7/73.0** | **96.5/81.0** | **97.5/86.2** | **97.3/89.3** | **97.8/94.5** | **98.4/96.5** | **95.1/85.8** | **96.7/91.3** | **97.9**/93.9 |

Table 16: Performance on different MRI modalities.

| No. Clicks | BRATS T1 | | | BRATS T1c | | | BRATS T2 | | |
|---|---|---|---|---|---|---|---|---|---|
| | 1 | 5 | 10 | 1 | 5 | 10 | 1 | 5 | 10 |
| ICNN* | 20.7 | 46.8 | 62.5 | 45.4 | 63.8 | 75.4 | 72.1 | 81.4 | 85.7 |
| Entropy-Min PI | 20.1 | 48.2 | 65.6 | 45.4 | 65.3 | 76.3 | 72.6 | 82.0 | 86.5 |
| Entropy-Min MI+PI | 13.8 | 38.1 | 57.1 | 42.3 | 57.8 | 66.8 | 70.0 | 81.2 | 86.8 |
| Cross-Entropy PI | 20.0 | 49.7 | 66.0 | 46.6 | 65.6 | 76.7 | 72.5 | 82.1 | 86.6 |
| Cross-Entropy MI+PI | 14.0 | 42.4 | 61.3 | 42.5 | 59.6 | 68.9 | 72.0 | 82.6 | 87.3 |
| IA+SA | 28.6 | 57.0 | 70.6 | 48.3 | 67.1 | 78.2 | 76.5 | 84.1 | 87.9 |
| TSCA | 34.4 | 60.9 | 74.0 | 50.1 | 70.2 | 79.6 | 77.7 | 85.8 | 89.0 |
| OAIMS(PI+MI) | **71.2** | **83.4** | **88.0** | **70.4** | **82.9** | **87.5** | **84.9** | **90.6** | **93.0** |

## A.10 EXPERIMENTS ON SOURCE-LIKE DATA

We here conducted additional experiments to evaluate our method's performance on source-like data, to ensure the adaptation does not degrade performance at the absence of distribution shift.

First, we tested the model directly on the exact domain it was trained on (source domain): Train on BraTS Flair (Train set) → Test on BraTS Flair (Test set). In this "no shift" scenario, we want to ensure that the adaptation mechanism does not degrade performance. As shown in Table 17, both our PI and PI+MI variants perform similarly to the non-adaptive baseline ICNN*, demonstrating the stability of our method.

Table 17: Performance on Source Domain (Minor Shift). Trained and Tested on BraTS Flair.

| No. Clicks | 1 | 5 | 10 |
|---|---|---|---|
| ICNN* | 93.4 | 95.6 | 96.4 |
| Ours (PI) | 93.4 | 95.5 | 96.3 |
| Ours (PI+MI) | 93.3 | 95.8 | 96.6 |

Second, we assessed a scenario to see if the model could regain performance on the source domain after adapting to a new domain. We first trained a model on BRATS Flair (train set), then adapted it to a different dataset (TBI Flair), and then applied it back to the original source domain (BraTS Flair). We compare this with performance of our backbone model (ICNN*) without adaptation. The results are shown in Table 18. When OAIMS adapts model parameters to TBI, the middle database, but then we do not allow it to re-adapt to the source data when it's tested on them (No Re-adaptation), it shows a slight drop in performance, especially with 1 click (90.5% vs 93.4%). However, if we allow OAIMS to re-adapt to the source data BRATS Flair via our standard online learning process (OAIMS (Re-adapt)), it recovers its original performance well, even with just 1 click of interaction.

We note that in practice, perhaps the most practical scenario are distribution shifts between training data and the database of deployment. Therefore, we are primarily interested in learning to adapt and perform well on the new domain, whereas continuing to perform well on the training domain is lower priority. However, this experiment shows the flexibility of OAIMS and its effectiveness to re-adapt to the original data easily and effectively.

Table 18: Performance on BraTS Flair after adapting to TBI.

| No. Clicks | 1 | 5 | 10 |
|---|---|---|---|
| ICNN* | 93.4 | 95.6 | 96.4 |
| OAIMS (No Re-adapt) | 90.5 | 94.8 | 95.7 |
| OAIMS (Re-adapt) | 93.5 | 95.8 | 96.6 |

## A.11 DATA EFFICIENCY OF OUR ONLINE ADAPTATION METHOD

To evaluate the data efficiency of our online adaptation method in unseen domains, we analyzed the number of samples required to reach satisfactory segmentation performance. For each target dataset, we partitioned the data into an adaptation split and a held-out evaluation split. Online adaptation was performed only on the adaptation split.

During the adaptation phase, the model was sequentially updated using images from the adaptation split, with 10 clicks per image. To track the efficiency of this process, we evaluated the model's performance on the held-out split after every five adaptation samples. During this evaluation step, we provided 10 simulated clicks for each image in the held-out split, without adaptating to them.

The results are visualized in Figure 7, where the x-axis represents the number of images processed for online adaptation, and the y-axis reports the average Dice score across all images in the held-out set. The left subfigure presents results of a model pre-trained on BraTS (FLAIR, T1, T1c) and adapted/evaluated on ATLAS, where the adaptation split and held-out split contain 554 and 100 images respectively. The red reference line indicates the *in-distribution* performance of an automatic baseline trained offline with standard supervised learning on the entire 554-image adaptation split, and evaluated on the held-out set from the same distribution. This gives a useful reference for what performance is meaningful to achieve by a model that was trained on a different distribution after online adaptation on the target distribution. The center subfigure shows results on the BraTS T1

test split using a model pre-trained on BraTS FLAIR. In this case the adaptation split contains 199 images and the held-out split contains 50 images. The right subfigure shows results on the WMH FLAIR using a model pre-trained on BraTS FLAIR. In this case the adaptation split contains 40 images and the held-out split contains 20 images.

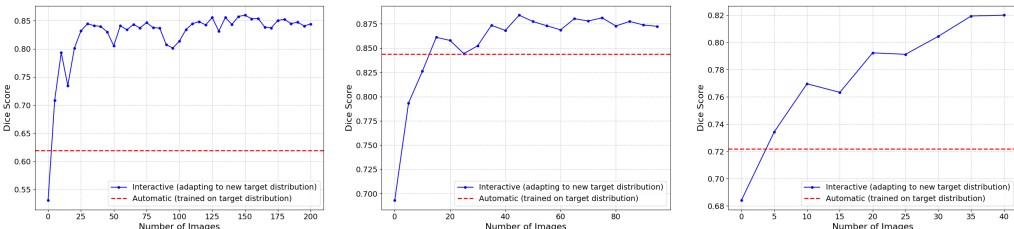

Figure 7: Online adaptation on ATLAS (left), BraTS T1 (center) and WMH FLAIR (right) datasets. The Dice score is calculated on corresponding held-out sets after every five adaptation samples. Online adaptation with our method reaches high performance to the new domain after processing only few (10-40) images, reducing the effort needed by the user for subsequent corrections.

As shown in Figure 7, the model exhibits high data efficiency. On the ATLAS dataset (left), the learning curve demonstrates a steep initial ascent; after adapting only to 5 images, the interactive model's performance on the previously unseen target distribution already surpasses that of the automatic baseline trained on this distribution. As the sequential adaptation progresses to approximately 30 images, the average Dice score stabilizes above 0.80. This represents a substantial gain over the pre-adaptation baseline score of <0.55 (at 0 images), clearly demonstrating the method's ability to quickly adapt to an unseen domain. The BraTS T1 results (center) show a similar trend: the interactive model outperforms the baseline after just 15 images and stabilizes at a higher performance level after 40 images. The WMH FLAIR results (right) also show that the interactive method quickly outperforms the baseline (after just 5 images) and achieves high performance with fewer than 40 cases.

## A.12    USE OF LLMS

We used a large language model (LLM) only to improve the writing. Specifically, the LLM was employed to revise some sentences, focusing on grammar and style. The LLM was not used for generating ideas, searching related work, or contributing to the scientific content of the paper.

