# OpenReview forum: "You Point, I Learn: Online Adaptation of Interactive Segmentation Models for Handling Distribution Shifts in Medical Imaging"
_ICLR.cc/2026/Conference — ICLR 2026 Poster_

### Official Review · Reviewer_oQJi · 2025-10-17

**Soundness:** 3
**Presentation:** 3
**Contribution:** 3
**Rating:** 6
**Confidence:** 3

**Summary:**

This paper presents an online adaptation framework for interactive medical image segmentation to address distribution shifts.
It introduces 2 types of adaptation: Post-Interaction and Mid-Interaction adaptation. A Click-Centered Gaussian loss is proposed to control how the model adapts to signals around user clicks.

**Strengths:**

The proposed framework, combining Post- and Mid-Interaction adaptation, along with the problem formulation, which integrates model updates with interactive segmentation, is interesting.

The CCG loss is a reasonable design choice and can effectively prevent the model from learning pixels' labels irrelevant to the click, focusing updates on relevant regions.

Experiments are comprehensive, covering fundus and brain MRI datasets. Experiment results are promising, demonstrating significant performance gains over existing methods, especially under large distribution shifts

**Weaknesses:**

It would be helpful if the authors could provide more implementation details on finetuning e.g., whether the entire U-Net is made trainable during adaptation or if some layers are frozen.

It would be helpful if the authors could explain why focal loss is used. Is the training data overwhelmed with easy-to-learn regions?

Mid-Interaction adaptation seems intimidating, because the user has to wait for the model to be updated before providing another click, which could introduce latency and disrupt workflow.

With only one gradient update step, it is possible that the updated model still predicts the wrong label for the user-clicked region. It would be helpful if the authors could provide some discussion.

The method's effectiveness relies on a reasonably good pseudo-ground-truth. In cases with very few clicks or extremely poor initial predictions, this assumption may not hold.

**Questions:**

Please refer to the weaknesses.

---

> ### Author Response · Authors · 2025-11-23
>
> Dear reviewer, thank you for your time and for offering constructive feedback\! We are glad that you acknowledged that “the CCG loss is a reasonable design choice,” which was also noted by reviewers MLug and n4wo. We also appreciate your recognition that “the experiments are comprehensive”,  and we are pleased that all reviewers gave positive assessments of the experiments.
>
> We **below address the points raised by the reviewer**:
>
> ---
>
> **\[W1\]: Details on finetuning**
>
> We clarify that all model parameters are updated during our online adaptation. We do not set any frozen layers, because that setting restricts the implementation of our OAIMS method to specific architectures. Even with fine-tuning on all parameters, our method performs better without showing observable overfitting, thanks to the robust design of our framework. **We added this clarification in section 2.4  and section 3.2 of the revised paper.**
>
> **\[W2\]: Why use focal loss:**
>
> Focal loss is widely used in segmentation and medical segmentation tasks because many parts of a segmentation mask are relatively “**easy**” (for example, the background or the middle of a tumor), while some parts (such as lesion borders) are much **harder**—a classical scenario of sample-difficulty imbalance. It is common to use it in combination with Dice loss or its variants in medical segmentation \[1, 2, 5, 6, 7\]. Multiple works **include experiments** showing that hybrids of Dice \+ Focal outperform using Dice alone \[1, 2, 3, 4\].  We changed the paper in section 2.2 to clarify why we used Focal loss .
>
> 1.Zhu W. *et al.* — *AnatomyNet: Deep Learning for Fast and Fully Automated Whole-volume Segmentation of Head and Neck Anatomy*
>
> 2.Gammoudi I. *et al.* — *HDFU-Net: An Improved Version of U-Net using a Hybrid Dice Focal Loss Function for Multi-modal Brain Tumor Image Segmentation*
>
> 3.Islam M. R. *et al.* — *Enhancing Semantic Segmentation with Adaptive Focal Loss: A Novel Approach*
>
> 4.Usman A. *et al.* — *Multimodal HIE Lesion Segmentation in Neonates: A Comparative Study of Loss Functions*
>
> 5.Zhou W. *et al.* — *EARDS: EfficientNet and Attention-based Residual Depth-wise Separable Convolution for Joint OD and OC Segmentation*
>
> 6.Yeung M. *et al.* — *Unified Focal Loss: Generalising Dice and Cross Entropy-based Losses to Handle Class Imbalanced Medical Image Segmentation*
>
> 7.Garcia-Salgado, B. P. *et al.* — *Enhanced Ischemic Stroke Lesion Segmentation in MRI Using Attention U-Net with Generalized Dice Focal Loss*
>
> **\[W3\]: Time latency for our work:**
>
> Efficiency is critical for this kind of work. Since Mid-Interaction (MI) updates the model after *each* click, any significant delay would make the tool not convenient for clinicians. This is exactly why we prioritized designing a relatively simple but effective method.
>
> We have measured the computational latency on the BRATS database and **added clarfication to the section 3.2 of the paper (Computational latency).** We provide the table here for the reviewer’s convenience:
>
> |  | MI (after each click) | PI (after image) |
> | :---- | :---- | :---- |
> | **GPU (A5000)** | 0.05s | 0.09s |
> | **CPU** | 0.25s | 0.41s |
>
> On a GPU (NVIDIA A500), the method is extremely fast, with updates taking nearly no time (0.05s) for MI. However, we understand that real clinical environments may not have access to such hardware. Therefore, we tested a practically applicable scenario: performing the online adaptation on a standard CPU. As shown, the wait time after each click is only **0.25s**. This is practically imperceptible to the user and ensures a smooth workflow. The Post-Interaction (PI) update is also very quick (0.41s). As a result, our method demonstrates strong clinical feasibility. This validates our design priority of creating a simple yet effective model specifically tailored for deployment in resource-constrained real-world environments.

---

> ### Author Response · Authors · 2025-11-23
>
> **\[W4\]: Can we do multiple gradient updates per iteration instead of one?**
>
> It is a trade-off. While multiple updates per interaction could potentially improve performance by learning more from a single step, they also increase the risk of overfitting to specific images or interactions. For instance, prior work like IA+SA updates the model multiple times per click but is forced to reset parameters after finishing clicking on one image to prevent this overfitting, which wastes computational resources. In contrast, using a single update generation is a safer strategy.
>
> Furthermore, computational efficiency is critical for online adaptation in interactive segmentation, where the system must respond very quickly. Although our method is fast, running multiple optimization steps for every point would introduce latency that hinders real-world clinical applicability. Therefore, our experiments aim to demonstrate that a single iteration can achieve effective adaptation while maintaining the speed required for deployment.
>
> In practice, the number of update steps could be treated as a hyperparameter and chosen for specific datasets or clinical needs to further improve performance. However, for this paper, we prioritized this "safe and efficient" setting to demonstrate that our method remains effective and outperforms other methods even with a single update step.
>
> **\[W5\] Very few clicks or extremely poor initial predictions:**
>
> Great point, and thank you for the opportunity to study this further.
> **We improved the manuscript by adding the below experiments to Section 3.3 of the main paper**.
>
> We conducted two types of experiments as stress tests to demonstrate the method works even when **initial predictions are highly erroneous (Scenarios A and B) or when only a small number of clicks are available (Scenarios A and Table 4 in the main paper) .**
>
> **Scenario A: Extreme Domain Shift (BraTS Flair → Atlas T1).** We trained the model *only* on BraTS Flair and tested it on Atlas T1. This represents a very large domain gap where initial predictions without online adaptation are extremely poor (starting \<10% Dice with 1 click). The results are shown in the below table.
>
> First, we examine performance when only 3 clicks are performed per image. Just with 1 localisation click and no online adaptation, the **base interactive model** (ICNN\*) achieves 9.3% Dice, very poor segmentation. After 3 clicks, it still performs very poorly (12.8% Dice), although it does improve slightly with each click, **indicating that the corrective functionality of the model after clicks is still functioning, but very slightly**. When we add our method’s online learning functionality to this base model, and use those very bad initial segmentations as pseudo-ground truth (pseudo-GT), the method shows significant improvement. For example, with 3 clicks, online learning leads up to 57.9% Dice. This demonstrates that the very poor initial pseudo-GT, of 12.8% with 3 clicks, does not destroy the online learning process.
> We repeat the experiment but allow 10 clicks per image. This simulates a more realistic scenario than the above, when the user uses more corrections to improve an initially extremely bad segmentation. Our method of online learning recovers a high 80% Dice score, which is remarkable considering it has to learn to adapt starting from very low quality pseudo-GT. We note that these improvements are much greater than the strongest prior method, TSCA.
> | Method | 1 click |  | 3 Clicks | 10 licks |
> | - | - | - | - | - |
> | **Base interactive model (No Adapt)** |9.3|  |12.8|24.3|
> | **TSCA** |9.3|  |40.7|65.6|
> | **OAIMS (Ours)** |25.6|  |57.9|80.0|
>
> **Scenario B: "Worst-First" Sequential Ordering.** We tested the extreme case where the model processes images in an order that presents the "hardest" images first (those where the model segments with lowest Dice% with one click first). The first images have almost 0% Dice (extremely bad initial prediction). Consistent with the experiments in the original paper, we applied a total of 10 clicks per image. As shown below, the performance on ATLAS is similar to performance achieved with random ordering, indicating that extreme initial model failures do not influence the model significantly.
> | Method |Click 1|Click 5|Click 10|
> | - | - | - | - |
> | **OAIMS  (Random Order)** |67.5|82.2|86.0|
> | **OAIMS (Worst-First)** |66.4|81.6|85.3|
>
> Beyond these two additional stress tests, **we already include experiments regarding limited user clicks in Table 4 of the main paper**. This table includes experiments with 3 clicks per image(1 initial \+ 2 corrections) and demonstrates that our method outperforms competing approaches.
>
> ---
>
> Thank you again for taking the time to review our work. We hope we have fully addressed all of your comments and that this strengthens your support for the publication of our article. If you have any further questions, please let us know.

---

### Official Review · Reviewer_n4wo · 2025-10-27

**Soundness:** 3
**Presentation:** 3
**Contribution:** 2
**Rating:** 6
**Confidence:** 5

**Summary:**

This paper proposes an online adaptation framework for interactive medical image segmentation under distribution shift. The method, termed OAIMS, introduces two adaptation mechanisms, Post-Interaction (PI) and Mid-Interaction (MI), that update model weights during and after user interaction sessions. The approach employs a Click-Centered Gaussian (CCG) loss to strengthen responsiveness around user clicks and to improve model generalization across unseen modalities or pathologies. The authors evaluate their method on multiple fundus and brain MRI datasets, demonstrating superior performance over existing online adaptation methods such as IA+SA and TSCA.

**Strengths:**

1. The paper tackles a practically important problem, handling distribution shifts in medical image segmentation, by combining interactive segmentation with online adaptation.
2. The proposed OAIMS framework, integrating Post-Interaction and Mid-Interaction updates, is conceptually clear and technically well-structured.
3. The Click-Centered Gaussian (CCG) loss is an intuitive and effective design that enhances model responsiveness around user inputs.
4. Experiments cover a wide range of datasets (five fundus and four brain MRI), demonstrating consistent cross-domain generalization.
5. Comprehensive ablation studies verify the contribution of each component and show the robustness of the method.
6. The paper is clearly written, well-organized, and supported by informative visualizations that make the method easy to follow.

**Weaknesses:**

1. The method is only validated on a U-Net backbone; generality to Transformer-based or large segmentation models (e.g., SAM) is not demonstrated.
2. The experiments lack a practical interaction-efficiency metric (e.g., number of clicks to reach 80% IoU / 80% Dice), which is important for interactive workflows.
3. The pseudo–ground-truth assumption is strong, and if the model’s initial predictions are highly erroneous, online updates may accumulate errors and cause distributional drift.
4. The baseline comparisons omit the latest (2025) interactive segmentation methods; the novelty claim for CCG is overstated given related prior work (e.g., methods that weight click neighborhoods or modify attention such as AdaptiveClick (TNNLS, 2025)).
5. Computational cost and latency of online updates are not reported, leaving clinical feasibility unclear.

**Questions:**

1. How computationally expensive is the online adaptation per image? Could this realistically run in real-time clinical use?
2. Have the authors tested their method with actual human annotators rather than simulated clicks?
3. How sensitive is the method to noisy or inconsistent initial inputs?
4. Could the pseudo–ground-truth mechanism lead to error accumulation (catastrophic drift) during long adaptation sequences?
5. How does OAIMS behave under minor distribution shifts, does continual adaptation degrade performance on source-like data?
6. Compared to strong non-online interactive models (e.g., nnInteractive), does online learning provide a clear advantage?
7. The method focuses on point clicks, can CCG also adapt to other interaction modalities such as bounding boxes or scribbles?
8. The paper claims PI adaptation provides benefits even when MI has already been applied; however, if MI is trained on one modality, can MI alone be applied on a new modality without retraining PI and still be effective?

This is a solid paper, and if the authors can address these questions, I would consider increasing my score.

---

> ### Author Response · Authors · 2025-11-23
>
> Dear reviewer, thank you for your effort in reviewing this work and the constructive feedback.
> We are glad the review acknowledges that the paper “tackles a practically **important problem**” and that the method is sound, stating that “the **proposed OAIMS** framework… **is conceptually clear and technically well-structured**.”
> It is satisfying to see that the evaluation is found convincing, with the review recognizing that “**comprehensive ablation studies verify the contribution** of each component and show the **robustness of the method**”.
> Gladly, the paper’s writing is also found to be “**clearly written, well-organized** “ that makes “the method **easy to follow**”.
>
> We thank the reviewer for the overall assessment where they comments that “**This is a solid paper**” which you would be happy to support further by increasing the score once your questions are addressed. We below **address all points raised:**
>
> (Apologies for the length of the response but we wanted to clarify all your questions clearly.)
>
> ---
>
> **\[W1\] : Generality to Transformer-based or large segmentation models:**
>
> The reviewer raises that the work does not evaluate whether the method will work with Transformer models (or large SAM models).
>
> First of all, we emphasize that the paper **already contains a comparison of a Convolutional Unet backbone** (ICNN\* in text/tables, interactive, non-adaptive) **with a SAM**\-based model, specifically MedSAM-Adapter (Med-SA in text/tables). In fact, this was one of the earliest explorations in our investigation, to determine what backbone to use. As shown in the results, our **CNN-based backbone method outperformed the SAM-based baseline**, particularly as the number of interaction clicks increased. This is consistent with recent literature that finds Convolutional nets to be equally good, or even better, than Transformer / ViT-based models \[1,2,3 below\]. Therefore we employed the more effective CNN backbone in the rest of the study.
>
> Furthermore, Transformer / ViTs (and variants, e.g. SwinUnet) are **slower** than similar size Convnets. Even worse, SAM models are **much larger** and more **computationally intensive, hence even slower** (not only training, which is slow even with Parameter-Efficient methods ala LoRA , but inference is also slower). In a practical online adaptation setting, where the model is supposed to be updated during use, ideally on standard commodity hardware of users (not high end research machines), the high computational cost of Transformers makes them less attractive. Even in research settings, the high computational cost makes experiments less practical, especially for labs with limited resources. Since our comparison of ConvNet and  MedSAM (which is ViT) showed no advantage of the latter Transformer, we found no benefit to further explore the latter.
>
> Methodologically, our method’s design is model agnostic. There is no theoretical barrier to applying our framework to Transformer-based architectures. Our online learning strategy is based on manipulation of output predictions and loss function optimization, not the internal layers/components of the network. However, for the reasons above, we strongly believe it is preferable **to focus our limited hardware resources on making a deeper analysis, with varied experiments to provide rich insights** (e.g. extensive ablation study, as acknowledged by the reviewer), rather than repeat similar experiments with other backbones.
>
> **We updated the paper, Section 2.2,** to state reasoning for choice of Unet and that the method is model agnostic and amenable to other architectures.
>
> References that show CNNs (and Unets) are (still) equally good (or better) than Transformer / ViT models:
>
> - Kazaj et al, From Claims to Evidence: A Unified Framework and Critical Analysis of CNN vs. Transformer vs. Mamba in Medical Image Segmentation, 2025
> - Isensee et al, nnU-Net Revisited: A Call for Rigorous Validation in 3D Medical Image Segmentation, MICCAI 2024
> - Liu et al, A ConvNet for the 2020s, CVPR 2022

---

> ### Author Response · Authors · 2025-11-23
>
> **\[W2\]:** **Assessing interaction-efficiency between methods (e.g. via “Number of clicks to target dice”):**
>
> Thank you for the suggestion. Although we do not explicitly measure this suggested metric, this aspect of method performance is already assessed in the original submission in a different manner. We reported performance of methods when a different numbers of clicks is performed on each image. For example, Table 1 and 2 report performance after 1, 5, 10 clicks. Table 4 compares our method with the prior method that performed best in our experiments, TSCA, when only 3 or 5 clicks are allowed per image. Our method performs significantly better than all baselines **at each setting** , even with limited clicks. This means **our method is more “interaction-efficient”** (higher performance with less clicks), which is what the reviewer enquired, but assessed in a different manner.
>
> Regardless,, we take the reviewer’s advice and performed a comparison with this metric, to ensure we arrive at the same conclusions regarding our method performance. For this, we performed an experiment on the BraTS database (Training on Flair, Testing on T1, T1c and T2 modalities separately, to assess performance under 3 types of distribution shift). We set a **target Dice of 80%**, with a maximum cap of 20 clicks (if the segmentation does not reach 80% Dice after 20 clicks, we record it as 20 clicks). We compared our model with the strongest baseline (TSCA). The results (number of clicks to reach 80% Dice) are presented below:
>
> | Method | T1 (80%) | T1c (80%) | T2 (80%) |
> | :---- | :---- | :---- | :---- |
> | **TSCA** | 10.60 | 8.71 | 3.63 |
> | **Ours (PI+MI)** | **4.43** | **4.59** | **2.33** |
>
> We see that our method requires almost half the number of clicks to reach 80% Dice across all modalities, providing further evidence about the methods strong performance.
>
> **To improve the manuscript, we added this experiment in Appendix A.9. Thank you.**

---

> ### Author Response · Authors · 2025-11-23
>
> **\[W3\] ,\[Q3\], \[Q4\]: Will errors accumulate in the case of low-quality pseudo ground truth (i.e. bad predictions) or noisy input (i.e. wrong clicks from user)?**
>
> Great point, and thank you for the opportunity to study this further.
> **We improved the manuscript by adding the below experiments to Section 3.3**.
>
> We conducted three types of experiments as stress tests to demonstrate the method works even when **initial predictions are highly erroneous ( A and B) or user inputs (clicks) are noisy (C).**
>
> **A: Extreme Domain Shift (BraTS Flair→  Atlas T1)**. We trained the model *only* on BraTS Flair and tested it on Atlas T1. This represents a very large domain gap where initial predictions without online adaptation are extremely poor (starting <10% Dice with 1 click).
>
> First, we examine performance when only 3 clicks are performed per image. Just with 1 localisation click and no online adaptation, the **base interactive model** (ICNN\*) achieves 9.3% Dice, very poor segmentation. After 3 clicks, it still performs very poorly (12.8%), although it does improve slightly with each click, **indicating that the corrective functionality of the model after clicks is still functioning, but slightly**. When we add our method to the model, and use those very bad initial segmentations as pseudo-ground truth (pseudo-GT), the method shows significant improvement. This demonstrates that the very poor initial pseudo-GT (12.8% with 3 clicks) does not destroy the online learning process.
> We repeat the experiment with 10 clicks. This simulates a more realistic scenario, when the user uses more corrections to improve an initially extremely bad segmentation.  Our method of online learning recovers a high 80% Dice. These improvements are much greater than the strongest prior method, TSCA.
>
> | Method | 1 click | 3 Clicks | 10 Clicks |
> | - | - | - | - |
> | **Base interactive model (No Adapt)** | 9.3 | 12.8 | 24.3 |
> | **TSCA** | 9.3 | 40.7 | 65.6 |
> | **OAIMS (Ours)** | **25.6** | **57.9** | **80.0** |
>
> **B: "Worst-First" Sequential Ordering.** We tested the extreme case where the model processes images in an order that presents the "hardest" images first (those where the model segments with lowest Dice% with one click). The first images have almost 0% Dice. This can help assess how much of an issue can error accumulation be. Consistent with the experiments in the original paper, we applied a total of 10 clicks per image. As shown below, the performance on ATLAS is similar to performance achieved with random ordering, indicating that extreme initial model failures do not influence the model significantly.
> | Method | Click 1 | Click 5 | Click 10 |
> | - | - | - | - |
> | **OAIMS  (Random Order)** | 67.5 | 82.2 | 86.0 |
> | **OAIMS (Worst-First)** | 66.4 | 81.6 | 85.3 |
>
> **C:  Robustness to Noisy/Wrong Clicks** To test robustness to noise in input clicks (e.g. where users make mistakes), we simulated "Wrong Clicks". We simulate wrong user clicks by generating clicks on areas that were predicted with the correct class by the model, but we assign the wrong class to the click (i.e. as if the user clicks on a correctly segmented area and says to the model this is the opposite, wrong class). We tested three sub-scenarios:
>
> * **40% Points Wrong:** 40% of all clicks provided are randomly incorrect.
> * **First 4 Points Wrong: The first 4 correction clicks for each image are incorrect.**
> * **40% Images Wrong:** 40% of the images receive only wrong clicks from the user.
>
> The table below compares performance of OAIMS against the non-adaptive baseline on BRATS T1. Despite the high noise, our method demonstrates remarkable resilience compared to the baseline (Under the same noise conditions).
> | Scenario | Method | Click 1 | Click 5 | Click 10 |
> | - | - | - | - | - |
> | **1\. 40% Points Wrong** | Baseline | 20.6 | 32.9 | 43.9 |
> |  | OAIMS | 67.8 | 71.8 | 74.6 |
> | **2\. First 4 Wrong** | Baseline | N/A | 13.3 | 43.3 |
> |  | OAIMS| N/A | 56.9 | 75.1 |
> | **3\. 40% Images Wrong** | Baseline | 21.0 | 40.4 | 53.8 |
> |  | OAIMS| 68.1 | 75.2 | 76.9 |
>
> **To provide you with an informal, intuitive explanation for this robustness against error-accumulation, we believe this is thanks to the Self-Correction via the "Human-in-the-Loop" nature of combining interactive-segmentation with online-learning:**  Unlike unsupervised methods, where errors accumulate unchecked, interactive segmentation is self-correcting. If an update introduces a new error (due to a bad pseudo-GT or wrong click), the click provided by the user in later steps may correct this. This iterative feedback loop continually "overwrites" bad updates with new, correct information, preventing the "snowball effect" of error accumulation. There are other advantages, specific to our method. With CCG Loss, the learning is concentrated on the region corrected by the click, preventing the model from overfitting to errors that occur in other areas of the prediction and remain uncorrected.

---

> ### Author Response · Authors · 2025-11-23
>
> **\[W4\], \[Q6\]: Comparisons with other recent (2025) interactive models and novelty:**
>
> First, while there are many (non-adaptive) interactive methods, such as AdaptiveClick and nnInteractive that the reviewer mentions, they are not the primary target for our comparison. Our work concentrates specifically on **online adaptation** within interactive models. Recent i**nteractive methods (like nnInteractive**, a non peer reviewed Arxiv preprint at the time of submission and this review) do not perform online adaptation; meaning they cannot utilize information from user prompts to update the model parameters and adapt to domain shifts. Therefore, they are not in the same category as our work. We primarily compare against online adaptation methods based on interactive segmentation, using as comparison the state-of-the-art TSCA and IA+SA methods. We are not aware of published 2025 online adaptive segmentation work that demonstrates convincingly superior performance than these – if you are aware of any please provide references and we could discuss further. Furthermore, theoretically, new interactive backbone architectures could simply serve as backbone for our online-learning framework, so their performance does not conflict with ours. Combining them could potentially yield even better real-world utility.
>
> AdaptiveClick (TNNLS 2025\) that the reviewer mentions, is not an online adaptation method. Instead, it focuses on handling click ambiguity (where a single click might imply different masks). The model generates multiple candidate masks with different queries and identifies the optimal one that matches the user's intent. This is a fundamentally different task. The reviewer questions the novelty in our work in comparison to this. We assume because AdaptiveClick contains a “click loss” which may seem related to CCG, but it is not. The "click loss” in Adaptive click is “The cross-entropy loss is used as the click loss, and…(it)... is intended to compute classification confidence for each click”. So, it has a completely different form than our CCG, that employs a Gaussian and a class-specific indicator (see end of Sec.2.3), both shown important aspects of  its design in our ablation studies. Its intended function (compute classification confidence and handle click ambiguity) differs entirely from CCG, which strengthens the model’s spatial response around clicks and emphasize the most valuable regions for adaptation.
>
> There is prior work that weights neighborhoods around clicks, such as by indicating a Gaussian neighbourhood as “input” (e.g. ICNN\*), mostly in non-adaptive interactive segmentation literature. However, we are not aware of a \*training loss\* with the form of CCG, which would be primarily relevant to **online adaptation** of interactive segmentation, where literature is limited.
>
> (The reviewer mentions nnInteractive, an ArxIv preprint that is unpublished, non-peer reviewed at the time of submission and this review (“Compared to strong non-online interactive models (e.g., nnInteractive)..”). The guidelines of ICLR clearly state that submissions are not expected to compare against non-published work, including Arxiv, and reviewers should not penalize for this. We mention above the difference with nnInteractive that is not an adaptive online learning, which could actually serve as backbone model for our online-adaptation framework. But, with all due respect, we will not compare against nnInteractive more directly as this would set a negative precedent for the reviewing process.)
>
> **\[W5\],\[Q1\]: Computational cost and latency:**
>
> Efficiency is critical for this kind of work. Since Mid-Interaction (MI) updates the model after *each* click, any significant delay would make the tool not convenient for clinicians. This is exactly why we prioritized designing a relatively simple but effective method.
>
> We have measured the computational latency on the BRATS database and **added clarfication to the section 3.2 of the paper (Computational latency).** We provide the table here for the reviewer’s convenience:
>
> |  | MI (after each click) | PI (after image) |
> | :---- | :---- | :---- |
> | **GPU (A5000)** | 0.05s | 0.09s |
> | **CPU** | 0.25s | 0.41s |
>
> On a GPU (NVIDIA A500), the method is extremely fast, with updates taking nearly no time (0.05s) for MI. However, we understand that real clinical environments may not have access to such hardware. Therefore, we tested a practically applicable scenario: performing the online adaptation on a standard CPU. As shown, the wait time after each click is only **0.25s**. This is practically imperceptible to the user and ensures a smooth workflow. The Post-Interaction (PI) update is also very quick (0.41s). As a result, our method demonstrates strong clinical feasibility. This validates our design priority of creating a simple yet effective model specifically tailored for deployment in resource-constrained real-world environments.

---

> ### Author Response · Authors · 2025-11-23
>
> **\[Q2\]: Test with human annotators:**
>
> We agree that studies with human users are valuable for validating interactive methods. Respectfully, however, conducting a truly informative user study, **especially in healthcare**, is **highly more complex and expensive** than it may appear. First, performing a user-study requires development of a Graphical Interface. It then requires recruiting radiologists with the specific relevant expertise (e.g., oncology), which is difficult given their limited availability, high cost, and limited accessibility to most labs. Moreover, the experiments require multiple users with adequate training. To compare with different methods, we need multiple annotations per method, and online adaptation adds considerable continued annotation. Therefore the organization and execution of healthcare studies can require several months of planning to ensure unbiased, informative results. This is infeasible for a revision and beyond the paper’s  scope. We also respectfully note that, for the same reasons, many prominent interactive segmentation methods, including those published in **top-tier conferences** (e.g. CVPR, ECCV) and journals, also rely on simulation-based protocols for evaluation and **usually do not contain user studies**. This includes the original publications of SOTA methods **(IA+SA and TSCA) we compare against.**
>
> We would also like to clarify that our simulation protocol was designed to be challenging to assess method robustness. Instead of always simulating clicks at the exact center of the largest erroneously segmented region (a deterministic, non-human-like behavior **used in many other interactive segmentation papers \[e.g. IA+SA, ITIS, IteR-MRL, and IL-TTOA\])**, our simulation randomly samples a point within the erroneous region. This assesses the method’s robustness under challenging variability of clicks. We believe the method’s strong performance across multiple, diverse datasets, provides convincing evidence of its effectiveness in comparison to SOTA (thankfully acknowledged by your review, “show the robustness of the method”, and other reviewers, e.g., “convincingly show that their method works better” by reviewer MLug) to the extent that it warrants the publication of the findings to the community.
>
> We do, however, acknowledge that it is meaningful to move towards enabling users to interact with the method. To address this, **we are developing an easy-to-use GUI** that integrates our method. We will **plan to release this GUI alongside our code** upon publication, allowing other researchers to easily and interactively test the method.
>
> ITIS: Iteratively Trained Interactive Segmentation (Mahadevan et al.)
>
> IteR-MRL: Iteratively-Refined Interactive 3D Medical Image Segmentation with Multi-Agent Reinforcement Learning (Liao et al.)
>
> IL-TTOA: Interactive Few-Shot Learning: Limited Supervision, Better Medical Image Segmentation (Feng et al.)

---

> ### Author Response · Authors · 2025-11-23
>
> **\[Q5\] Does performance of OAIMS (continual adaptation) degrade on source-like data (i.e. with minor distribution shift)?**
>
> Fair question, as some methods may regularize optimization to handle data from other domains, but the same regularizer could make optimization suboptimal for source-like data. There is no such aspect in the design of our method. In fact, the smaller the shift, the more probable it is to obtain better model prediction, thus better pseudo ground truth, and consequently optimization should be even easier without issues.
>
> We performed **new experiments** to assess this, which we **added in Appendix A.11 of the revised paper to improve it:.**
>
> **1\.** First, to evaluate the performance between minor distribution shift, we tested the model directly on the exact domain it was trained on: Train on BraTS Flair (Train set) → Test on BraTS Flair (Test set). In this "no shift" scenario, we want to ensure that the adaptation mechanism does not degrade performance. As shown in the following **Table**, both our **PI** and **PI+MI** variants perform similarly to the non-adaptive baseline ICNN\*, demonstrating the stability of our method.
>
> | Method | Click 1 | Click 5 | Click 10 |
> | :---- | :---- | :---- | :---- |
> | **ICNN\*** | 93.4 | 95.6 | 96.4 |
> | **Ours (PI)** | 93.4 | 95.5 | 96.3 |
> | **Ours (PI+MI)** | 93.3 | 95.8 | 96.7 |
>
> 2\.  Second, we assessed a scenario to see if the model could regain performance on the source domain after adapting to a new domain. We first trained a model on BRATS Flair (train set), then adapted it to a different dataset (TBI Flair), and then applied it back to the original source domain (BraTS Flair). We compare this with performance of our backbone model (ICNN\*) without adaptation. When OAIMS adapts model parameters to TBI, the middle database, but then we do not allow it to re-adapt to the source data when it’s tested on them (No Re-adaptation), it shows a slight drop in performance, especially with 1 click (90.5% vs 93.4%). However, if we allow OAIMS to re-adapt to the source data BRATS Flair via our standard online learning process (OAIMS (Re-adapt)), it recovers its original performance well, even with just 1 click of interaction.
>
> We note that in practice, perhaps the most practical scenario are distribution shifts between training data and the database of deployment. Therefore, we are primarily interested in learning to adapt and perform well on the new domain, whereas continuing to perform well on the training domain is lower priority. However, this experiment shows the flexibility of OAIMS and its effectiveness to re-adapt to the original data easily and effectively.
>
> | Method | Click 1 |  Click 5 |  Click 10 |
> | :---- | :---- | :---- | :---- |
> | ICNN\* | 93.4 | 96.5 | 96.4 |
> | OAIMS (No Re-adapt) | 90.5 | 94.8 | 95.7 |
> | OAIMS (Re-adapt) | **93.5** | **95.8** | **96.6** |
>
>
> **\[Q7\] Other types of prompts BBox/Scribble**
>
> Absolutely, in fact we are looking into it for future extensions. Most parts of the method (components and steps in MI and PI) are generic. The only aspect that is primarily click-specific currently is CCG. If one introduces losses that enforce the model to learn the areas defined by BBoxes and Scribbles in Eq. 5, these can easily be integrated. Even CCG could be adapted to boxes and scribbles. Intuitively, extending CCG to a scribble is likely easy: Apply Gaussian over all pixels of scrible. For a bounding box (bbox), it may not be as intuitive, but a draft idea is to force the model to correctly segment the target class *inside* the bbox, while penalizing target predictions that appear *outside* the box and close to the box boundaries.
>
> We updated the conclusion (Sec. 4\) to comment on this in the revised article.
>
>  **\[Q8\]: “can MI alone be applied on a new modality without retraining PI“?**
>
> We interpret this question as asking whether **Mid-Interaction (MI)** can be used effectively on its own, without Post-Interaction (PI).
>
> The original manuscript includes an experiment in the ablation study (**Table 6**) that addresses this. The results show that MI alone provides a performance benefit compared to the baseline. However, the non-adaptive baseline was not originally included in that table, so this is what likely did not allow the reviewer to make a comparison (baseline VS MI only) to see that MI alone offers benefits. **We improve the revised manuscript by adding this baseline row to Table 6, to make such insight clear.** Thank you.
>
> ---
>
> Thank you once again for your reviewing. We are very glad you acknowledge this is a “solid paper”. We hope that this addresses all the main points that affect your decision, so that you can support the publication of the article even further by increasing the score. If you have any further important questions, please let us know.

---

### Official Review · Reviewer_MLug · 2025-10-28

**Soundness:** 3
**Presentation:** 3
**Contribution:** 3
**Rating:** 8
**Confidence:** 4

**Summary:**

This paper proposes an online learning technique for interactive image segmentation in the medical domain, where they specifically propose it for domain adaptation (which is one of the main problems in medical imaging). In particular, the main innovation seems to be a Click-Centered Gaussian loss function, which is a loss which acts on interactive clicks and makes the model react more quickly to user-clicks than previous methods. This loss is applied in a pre-training phase (out-of-domain pre-training phase), a Mid-interaction phase (i.e. between user-clicks in a single image), and a Post-interaction phase (i.e. after a single image is segmented). A well designed set of experiments show that the proposed loss is important in all three training phases, that their effects are cumulative, and that their final method outperforms the state-of-the-art.

**Strengths:**

* The Click-Centered Gaussian loss function is simple: It takes a user click, does a Gaussian convolution, and uses this as weights for a cross-entropy loss.
* The CCG loss makes sense: by focusing only on the area around the user click, it encourages the model to quickly respond to user interactions. In contrast to previous user-click losses which only operate on a single pixel, the Gaussian ensures that a larger portion of the image is taken into account which experiments demonstrate to be beneficial. Note that I prefer simple and effective methods, since these are more likely to get actual adoption in the field.
* The authors compare with two previous methods (IA+SA and TSCA) on six different datasets and convincingly show that their method works better.
* Stage two (page 4) of fine-tuning with multiple correction points is well explained how it avoids trivial network updates.
* There are a good amount of ablation studies to better understand the behavior of the proposed method.

**Weaknesses:**

* There is not a single study with a real human: typically a simulated user behaves differently from a real user. The paper would improve if there is at least on real human study on a single dataset where the authors show their method works. This would also entail having a variable number of user interactions during annotation. Ideally this should be compared to a similar experiment using the best competitive method TSCA.
* The sigma of the Gaussian is not ablated. Theoretically, a sigma close to zero goes back to having the loss only on a single point. A sweep over sigmas could help understand how much the ‘Gaussian’ part of the loss really contributes.
* There seems to be a contradiction in the writing: In stage 2 the authors argue that it is not possible to reuse the user’s original correction clicks to update the model as the model already yielded the final prediction using those corrections, leading to trivial updates. Then in stage 3 they seem to do exactly that: using the last click c_t to reinforce the resulting prediction p_t^initial. While I think both are valid, the way it is written now is confusing since it directly contradicts the previous paragraph. Also, it took me quite some time to figure out why p_t^initial has the ^initial modifier. Some careful rephrasing in both stage 2 and 3 are needed to solve this problem, but this should be easily doable.
* Mid-interaction and post-interaction learning are not new (e.g. IA+SA). But this method works better.
* Using Gaussians for points is also not new, but it is shown to be very effective in this context.

**Questions:**

* Is it possible to ablate the Sigma for the Gaussian points?
* Is it possible to do a small user study as described above?

---

> ### Author Response · Authors · 2025-11-23
>
> Dear reviewer, thank you for the time taken to review this study and the useful feedback. First of all, we  are glad the reviewer acknowledges as strength that the method is not complex and explicitly states that they **“prefer simple and effective methods, since these are more likely to get actual adoption in the field.”**. Indeed, simplicity was one of our primary targets when designing the method. Especially for methods that aim to alter (adapt) the model after deployment, at the hands of the user without oversight by ML experts, simplicity is key to reliability. Also, the method must remain computationally efficient to adapt quickly at the hands of the user and commodity hardware, therefore simple but effective solutions are needed. We also thank the reviewer for acknowledging that the study presents a **“well designed set of experiments”** with extensive evaluation “on **six different datasets** and **convincingly** show that their method works better””, and “method **outperforms the state-of-the-art.**”. Finally,  we appreciate the reviewer, like other reviewers, acknowledges the presentation as good (presentation score: 3), that the method “makes sense”, and that we provide “a good amount of ablation studies”, across main paper and appendix,,  that help understand the method’s behavior.
>
> We **below address the points raised by the reviewer** in order:
>
> ---
>
>  **\[W1\] and \[Q2\]: Human study would be useful**
>
> We agree that studies with human users are valuable for validating interactive methods. Respectfully, however, we suspect the reviewer may be underestimating the complexity of performing such a study (“There is not a single study with a real human”). Conducting a truly informative user study, **especially in healthcare**, is **highly more complex and expensive** than it may appear. First, performing a user-study requires development of a Graphical Interface. It then requires recruiting radiologists with the specific relevant expertise (e.g., oncology), which is difficult given their limited availability, high cost, and limited accessibility to most labs. Moreover, the experiments require multiple users with adequate training (e.g. as the reviewer suggests, use a specified, variable number of clicks). To compare with different methods, we need multiple annotations per method, and online adaptation adds considerable continued annotation. Because the organization and execution of healthcare studies require months of planning to ensure unbiased, informative results, unlike comparable computer-vision studies. This is infeasible for a revision and beyond our scope. We also respectfully note that, for the same reasons, many prominent interactive segmentation methods, including those published in **top-tier conferences** (e.g., CVPR, ECCV) and journals, also rely on simulation-based protocols for evaluation and **usually do not contain user studies**. This includes the original publications of SOTA methods **(IA+SA and TSCA) we compare against.**
>
> We would also like to clarify that our simulation protocol was designed to be challenging to assess method robustness. Instead of always simulating clicks at the exact center of the largest erroneously segmented region (a deterministic, non-human-like behavior **used in many other interactive segmentation papers \[e.g. IA+SA, ITIS, IteR-MRL, and IL-TTOA\])**, our simulation randomly samples a point within the erroneous region. This assesses the method’s robustness under challenging variability of clicks. We believe the method’s strong performance across multiple, diverse datasets, provides convincing evidence of its effectiveness in comparison to SOTA (gladly acknowledged in your review, “convincingly show that their method works better”, and similarly Reviewers n4wo and oQJi) to the extent that it warrants the publication of the findings to the community.
>
> We do, however, acknowledge that it is meaningful to move towards enabling users to interact with the method. To address this, **we are developing an easy-to-use GUI** that integrates our method. We will **plan to release this GUI alongside our code** upon publication, allowing other researchers to easily and interactively test the method.
>
>
> ITIS: Iteratively Trained Interactive Segmentation (Mahadevan et al.)
>
> IteR-MRL: Iteratively-Refined Interactive 3D Medical Image Segmentation with Multi-Agent Reinforcement Learning (Liao et al.)
>
> IL-TTOA: Interactive Few-Shot Learning: Limited Supervision, Better Medical Image Segmentation (Feng et al.)

---

> ### Author Response · Authors · 2025-11-23
>
> **\[W2\] and \[Q1\]: Misunderstanding that “the sigma of the Gaussian is not ablated” and its further analysis**
>
> We agree ablation of sigma is meaningful.The original submission **already** included related experiments in **Appendix A.3, Table 12** (hyperparameter tuning, including σ), because of limited space in the main paper, though we understand it may have been missed by the reviewer. The last three rows of the table show the average Dice score for a given σ (1, 3, 5). We can see that a relatively small σ reduces the method’s performance by \~7% Dice in BRATS, which confirms that using the Gaussian is useful over just optimizing for the central point (as the reviewer suggested).
>
> The existing experiments of Table 12 varied sigma along with other hyper-parameters. When reporting results for a value of sigma, the effect of other hyper-parameters was “marginalised out” by averaging-away the different values of other hyper-parameters (explanation in Tab.12). Motivated by the reviewer's suggestion and acknowledging that sigma is of particular interest, **we conducted another new ablation study, focused only on sigma** using BraTS T1, where we fixed all other hyperparameters and varied only σ. We tested σ∈{0,1,2,3,4,5}, where the σ=0 case was implemented by applying the CCG loss only on a single pixel, as reviewer requested. We add the new results in Appendix A.8. We also show the results below in table-format for convenience of the reviewer.
>
> | Sigma (σ) | 0 | 1 | 2 | 3 | 4 | 5 |
> | :---- | :---- | :---- | :---- | :---- | :---- | :---- |
> | **Dice (%) with 10 points** |  Unstable (highest: 83.2) *| 85.5 | 87.3 | 88.1 | 88.0 | 88.3 |
>
> As σ increases from 0 to 3, the performance improves significantly, as the model receives more information from surrounding pixels. When σ is large enough (e.g., σ≥3) the method seems robust to the choice of its value, achieving stable performance. We also observed that for σ=0 the adaptation was highly unstable, sometimes resulting in very low performance. For example, over 3 runs (RNG seeds) of the same experiment, the final score was 79.9, 46.9, and 83.2. This instability was only observed for σ=0. This is likely because the model can easily overfit a single pixel, which can strongly hinder performance.
>
> **We enhance the manuscript by adding the new results in Appendix A.8, Fig.6. We also added a reference to it in the ablation study within Section 3.2 of the main paper, to ensure future interested readers will not miss it.**

---

> ### Author Response · Authors · 2025-11-23
>
> **\[W3\]: Contradiction about the use of original user clicks for adaptation Mid-interaction and Stage-2 of Post-interaction:**
>
> The reviewer raised that there may be a contradiction in the method’s design or text. Although there is no contradiction, as we clarify below, this point seems to have caused a slight confusion to the reviewer. Therefore we agree this warrants further clarification to avoid confusing any future readers. As the reviewer also states in their review, this is rather easy to address via small modifications of the text.
>
> First of all, we clarify and remind to the reviewer:
>
> In Stage 2 of Post-Interaction: We assume the user has already provided a set $C\_{T}$ of T corrective clicks, which result to the model predicting a corrected segmentation $P\_{\text{final}} \= P\_{T} \= f(I, C\_{T}, \\theta)$. After user corrections, this is assumed to be high quality, and is used as pseudo-ground truth in Post-Interaction Stage 2\. For PI adaptation, we create a new set of artificial clicks, $\\hat{C}$, to make a new model prediction $\\hat{P}$. We then update the model via losses that compare $\\hat{P}$ with the pseudo-ground truth $P\_{\text{final}}$. We need artificial clicks, because it is not effective to reuse the original *set $C\_{T}$* of user clicks to make a prediction, as it will simply result in reproducing $P\_{\text{final}}$ the pseudo-label, hence loss=0 and there will be trivial updates.
>
> In Mid-interaction the process is different: Assume that after t-1 user clicks $C\_{t-1}$, we obtain prediction $P\_{t-1}=f(I, C\_{t-1}, \\theta\_{t-1})$. Then, a new user click c\_t further corrects it, hence we obtain corrected prediction $P\_{t}^{\text{initial}}=f(I, C\_T, \\theta\_{t-1})$. We then use our losses to penalize deviation of $P\_{t-1} $(non-corrected prediction of model) from the corrected $P\_t^{\text{initial}}$ (which serves as pseudo-ground truth for this step t). There is no issue in using the user-clicks here, because different clicks are used to generate $P\_{t-1}$ and $P\_t^{\text{initial}}$ (hence they are different, $loss\neq0$, and no trivial updates). The extra click provides new information from the user, improving $P\_t^{initial}$ over $P\_{t-1}$, hence the former is used as pseudo-label in the losses, while the latter as prediction that should be optimized. This leads to obtaining improved parameters $\\theta\_t$. Finally, the updated model reprocesses image $I$ and clicks $C\_{T}$, to obtain the refined output for this iteration, $P\_t \= f(I, C\_T, \\theta\_t)$ (notice $\\theta\_t$ the updated parameters, vs $\\theta\_{t-1}$ used for$ P\_{t}^{\text{initial}}$). This improved $P\_t$ is then considered the output of this iteration t, and is shown to the user, for further corrections in the next iteration t+1. We hope this clarifies both the use of artificial vs user clicks, and the difference between $P\_t^{\text{initial}}$ vs $P\_{t}$.

---

> ### Author Response · Authors · 2025-11-23
>
> **\[W4\]: Previous works related to PI and MI adaptation:**
>
> The general concept of updating the model to be optimal for the specific image, and after all corrections have been completed so that it improves for the rest of the sequence, has been explored by other works. As the reviewer notes, such method is IA+SA, which we already acknowledge in Sec.1, L91, “which combines…image-level… sequence-level”.
> Our work does not claim to invent these underlying concepts, but instead a new method/approach to combine them (L100, L108). Importantly, an important contribution of our work is presenting a method that is simple and effective, combining well-motivated components that complement each other elegantly, and demonstrably “works better” (as reviewer acknowledges) than state-of-the-art across multiple tasks. It is very important, as community, to have clarity on what are the fundamental and (ideally) simple components necessary for a task (here online adaptation), to avoid overwhelming ourselves with over-complex solutions (that often are not real advancements).This work helps with this, and we believe it is valuable to communicate this elegant and effective “recipe” to the community, which can inform better design of future works.
>
> Finally, we would like to clarify that there are various important design differences from other works. The devil is often in the details, which often make big difference. Specifically, for the example of  IA+SA that the reviewer notes, **some design differences are**:
>
> * Proposing the CCG loss for strengthening the model's response to clicks, in comparison to IA+SA that optimized CE for just the clicked pixels. Ablation shows this makes a big difference. (details about CCG design in next question/answer).
> * Our work shows that it is sufficient to perform a single SGD update for each adaptation step to obtain great results. IA+SA and other methods (possibly following common influences?) apply multiple SGD updates (10 in IA+SA). This can lead to overfitting and longer processing times.
> * By design, IA+SA in Mid-Interaction overfits the clicked pixel (cause of its loss) and image (multiple SGD iterations). Therefore, it discards the model weights obtained via MI, before it performs PI adaptation and processes the next image of the sequence.
> * For the same reason (overfit), IA+SA needs a further regularizer to control for it. In our method this is avoided thanks to the above differences, reducing complexity of the framework.
> * Other design differences including our two-stage PI process, the way pseudo-ground-truth is treated, and applying the loss on $P_{t}^{\\text{initial}}$​ and $P_{t-1}$.
>
>  **\[W5\]: Use of Gaussian around a point may not be new:**
>
> While the Gaussian kernel is often used to model a neighbourhood around a pixel, to our knowledge, we are the first to **derive a training loss of such Gaussian form** for **online adaptation in interactive segmentation**. In the plethora of works on (non-adaptive) interactive segmentation, Gaussian kernels are often used there to convey information around a click to a static, pretrained model, e.g. a gaussian around the click **as input** in our baseline ICNN \[Sakinis 2019\], but we are not aware of a **loss** with CCG’s form. Contrary, works on online **adaptation** via interactive segmentation are limited (even though the two match so elegantly\!) and we aren’t aware of a similar loss here. If the reviewer would provide a reference, we could discuss this further.
>
> Very importantly, while CCG looks very simple at a glance, it has design details that matter. As discussed in the last paragraph of Sec 2.3 (“Why not apply the loss to all surrounding pixels?”), a key component is that the loss is class-specific: it focuses only on the target class of the clicked pixel, not the whole surrounding region. The ablation study in Section 3.2, Table 7 shows this makes a significant difference.
>
> **We improved the writing in Sec 2.3 to further emphasize this** important aspect that differentiates CCG, by moving the sentences that explain this after Eq. 3 (they were at beginning of Sec 2.3), as this may help the reader to better grasp this point.
>
> ---
> We thank the reviewer for the meticulous and fair review. We believe we have addressed all points which we hope will reinforce their support for the paper\! If you have further questions, please let us know.

---

> > ### Comment · Reviewer_MLug · 2025-11-28
> > **I maintain my rating: 8, good paper**
> >
> > After carefully reading the other reviews and the rebuttal, I maintain my rating.
> >
> > Two other reviewers were already mildly positive and many points raised of all reviewers are sufficiently addressed. I also appreciate the authors response that real user studies are especially expensive in the medical domain.

---

> > > ### Author Response · Authors · 2025-11-28
> > > **"I maintain my rating: 8, good paper" => Thank you to reviewer**
> > >
> > > Dear Reviewer,
> > >
> > > Thanks a lot for all your work on the reviewing process.
> > >
> > > "After carefully reading the other reviews and the rebuttal, I maintain my rating.":
> > > We are glad that after reviewing holistically all reviews and rebuttals you are happy to keep supporting the paper. Sincerely, thank you for the detailed work and the effort you put into this.
> > >
> > > "Two other reviewers were already mildly positive and many points raised of all reviewers are sufficiently addressed.":
> > > Thank you for acknowleding it, a lot of work went into the revision to address sufficiently all points. We will keep revising the paper and code as optimally as possible towards the publication if accepted.
> > >
> > > "I also appreciate the authors response that real user studies are especially expensive in the medical domain.":
> > > Indeed it is, and we are glad you acknowledge it.
> > >
> > > Once again, thanks a lot for your effort, the constructive review, and your support.

---

### Official Review · Reviewer_KBrJ · 2025-11-02

**Soundness:** 2
**Presentation:** 2
**Contribution:** 1
**Rating:** 4
**Confidence:** 3

**Summary:**

This paper addresses distribution shifts in medical image segmentation by using interactive segmentation as an online adaptation mechanism. The authors propose a framework where user click inputs not only guide predictions but also serve as signals for adapting model parameters during deployment. The method consists of two adaptation components: Post-Interaction adaptation (updating after complete image refinement) and Mid-Interaction adaptation (updating after each individual click), both incorporating a Click-Centered Gaussian loss that enhances the model's responsiveness to user guidance. They state that the user-corrected outputs can be treated as pseudo-ground-truth labels, enabling the model to continuously learn from sequential test images without additional manual annotations. The framework strengthens click responsiveness during initial training and focuses learning on clinically relevant regions indicated by user interactions. Comprehensive experiments across various medical imaging datasets demonstrate consistent improvements over existing methods in handling distribution shifts, including unseen imaging modalities and novel pathologies. This work aims to provide a practical solution for deploying segmentation models in medical imaging environments where domain shifts are common and user interaction is already an integral part of the clinical workflow.

**Strengths:**

The paper’s direction is interesting, and treating the post‑interaction, user‑refined model as a pseudo ground truth is a thoughtful design choice.

The evaluation spans multiple levels of distribution shift and reflects realistic clinical scenarios in medical imaging.

**Weaknesses:**

The submission would benefit from greater technical depth, as the current contribution focuses primarily on the overall framework and the click‑centered Gaussian loss, which, on its own, is not fully convincing.

Moreover, I attempted to explore the advantages of the proposed method further. I have been trying to consider whether simply using an unsupervised loss based on the user’s inputs during inference would have improved the performance.

**Questions:**

The role and necessity of multiple correction points are unclear, especially if Stage‑1 user clicks are already assumed to be pseudo ground truth.

The introduction of artificial clicks risks injecting label noise during re‑training under the stated loss functions, and this potential degradation should be analyzed and controlled.

The click‑centered Gaussian loss lacks sufficient technical detail, and the paper does not provide empirical insights or ablations to explain its behavior and sensitivity.

The use of pretraining and its fidelity to real deployment are under‑specified, including whether the source model was trained with extensive click supervision and which loss functions were used.

A concise algorithmic description for both source training and test‑time inference with precise mathematical notation would significantly improve reproducibility.

Extending results to adaptation‑oriented segmentation benchmarks would strengthen claims of generality and applicability.

In Table 5 for BraTS, the inclusion of CCFL yields a large jump from 46.6 to 81.9, whereas similar gains are not observed elsewhere and are only marginal on BraTS T2, which warrants deeper analysis.

Two additional baselines would strengthen the study, namely standard cross‑entropy and entropy‑minimization losses applied at test time with pseudo labels or model predictions.

Clarification is needed on which network layers are updated during adaptation, given the combined image‑and‑click inputs and their contribution to re‑training effectiveness.

---

> ### Author Response · Authors · 2025-11-23
>
> Dear reviewer, thank you for your time and effort in reviewing our study. First of all, we are grateful that you acknowledge “the paper's **direction is interesting**.” and the stated strength that the “**evaluation spans multiple levels of distribution shift** and reflects r**ealistic clinical scenarios** “. In your summary you also state  “**comprehensive experiments** across various medical imaging datasets demonstrate **consistent improvements** over existing methods'', which was unfortunate this was not acknowledged as a strength that affects your scoring, unlike other reviewers that explicitly stated our extensive experiments are convincing that the proposed method improves over existing compared methods.
>
>
>
> The reviewer raised **2 weaknesses**, and **9 further questions**. We address all points below and hope that the reviewer reconsiders the strengths and weaknesses of the work accordingly, to support the publication of the study.
>
> ---
>
> **\[Weakness 1\]: “The submission would benefit from greater technical depth, as the current contribution focuses primarily on the overall framework and the click centered Gaussian loss, which, on its own, is not fully convincing.”**
>
> The work presents a framework with various components (Fig1) besides the CCG loss. It includes a method to perform MI (non trivial, see revised Sec. 2.4 with our added clarifications), a process for PI with 2 stages (one based on localisation clicks, 1 based on correction clicks), specific methodology for leveraging pseudolabels to derive losses and generate localisation/correction clicks, and a novel loss (class-restricted gaussian kernel, not to miss the class-restriction being important in our ablation studies). Each component may look simple on its own, but there is scientific value in determining and designing effective components that tie elegantly in an effective framework.
>
> The reviewer may feel that the method needs more technical depth because it appears relatively simple. However, achieving simplicity in science is a virtue, not a weakness. It is part of scientific progress to identify and distill the necessary effective components and avoid over-complication. From the beginning of the study, we deliberately strived to design a method that is as simple as possible, while being effective. For online adaptation of interactive methods to be practically useful, it needs to be reliable and fast (to adapt during user interaction and on commodity hardware). Our method’s simplicity has the indirect benefits of quick computation (simple losses based on pseudolabels and CCG that require little computation – we added discussion on computation in Sec. 3.2) and robust (simple SGD updates that avoid overfitting). At the same time, our extensive experiments show it is very effective (works across different distribution shifts, different number of clicks, different quality of predictions/pseudolabels…).
>
> The method’s effective simplicity is appreciated explicitly as Strength of the work by Reviewer MLug, who stated that they “**prefer simple and effective methods, since these are more likely to get actual adoption in the field.**”, and also agreed that the method “**makes sense**”, **which is consistent with the comments from the other 2 reviewers**. (“is an intuitive and effective design”,”is a reasonable design choice”). Another reviewer (Reviewer n4wo) noted that the overall framework is “conceptually clear and technically well-structured.” Many of the most successful techniques in machine learning are simple—for example, SGD, focal loss, Dice loss, Adam —because effective optimization needs to be computationally efficient and generalizable. Increasing complexity often introduces unnecessary computational burden and a higher risk of overfitting to specific tasks or datasets, rather than genuine generalizable advances.
>
> We evaluated our design with a deep analysis, via multiple ablation studies, which was acknowledged by reviewers: Reviewer MLug  noted that “there are a good amount of ablation studies to better understand the behavior of the proposed method,” and Reviewer n4wo commented that “comprehensive ablation studies verify the contribution of each component and show the robustness of the method. We also provide detailed information of the ablation study on the CCG loss in \[Q3\]. Therefore, given these ablation studies and the reviewers’ feedback, we hope that the reviewer may reconsider their view that the depth and value of the work is insufficient and not convincing.

---

> ### Author Response · Authors · 2025-11-23
>
> **\[W2\] “I attempted to explore the advantages of the proposed method further. I have been trying to consider whether simply using an unsupervised loss based on the user’s inputs during inference would have improved the performance.”**
> **and**
> **\[Q8\]: “Two additional baselines would strengthen the study, namely standard cross entropy and entropy minimization losses applied at test time with pseudo labels or model predictions.”**
>
> The second Weakness raised by the reviewer is that no unsupervised losses were tried, such as CE or Entropy Minimization with pseudolabels. These are among the first and most fundamental unsupervised losses. Many recent works are directly influenced by these ideas and extend them (including ours that uses pseudo labels). Therefore our comparisons focused on state-of-the-art methods that directly leverage user corrections. However, we do acknowledge that re-evaluating such methods is meaningful and thus take the reviewer’s advice and add new experiments with unsupervised methods.
>
> We implemented CE and Entr.Min with pseudolabels, and evaluate them with **new experiments on the Fundus and the BRATS datasets**. We use these unsupervised losses instead of our own methodology within the MI and PI steps. We study 2 settings for both losses, PI alone (i.e. optimization with target the pseudolabel after all user corrections were completed), and MI+PI (includes optimization after each click). Note: We ran experiments to configure optimal hyper-parameters per unsupervised method to get optimal performance from them, shown below.)
>
> As shown in **table below for Fundus imaging,** these losses provide almost no improvement and sometimes even yield worse performance than our non-adaptive backbone model (ICNN\*). This is likely because the information gained from unsupervised losses is limited. **Our method, OAIMS, outperforms consistently,** especially with less clicks.
>
> When the distribution shift is larger, between **BRATS datasets,** **Online learning methods such as  IA+SA and our OAIMS outperform unsupervised losses in all cases with much greater difference (\~5-60% Dice dependent on setting)**, especially on BRATS T1 and T1c.
>
> These experiments show that these unsupervised losses by themselves are not enough, and instead it is valueable to design elegant frameworks that effectively leverage signal from user interactions to adapt to new data, like our OAIMS framework.
> We believe these should **re-assure the reviewer that the method is effective** for addressing this very practical interactive-segmentation workflow. **We enhance the manuscript by adding the new results in Appendix A.10.**
>
> **(A) Performance on Fundus Imaging (Disc/Cup Dice %)**
>
> | Method |                   G1020 |  |  |                    PAPILA |  |  |                       GS1 |  |  |                GAMMA |  |  |
> | :--- | :--- | :--- | :--- | :--- | :--- | :---- | :---- | :---- | :---- | :---- | :---- | :---- |
> | No. Clicks | 1 | 5 | 10 | 1 | 5 | 10 | 1 | 5 | 10 | 1 | 5 | 10 |
> | ICNN\* | 89.4/77.4 | 93.1/83.1 | 95.1/88.0 | 88.3/60.3 | 92.6/70.2 | 94.6/77.1 | 96.6/82.4 | 97.2/90.4 | 97.7/94.1 | 94.4/83.3 | 96.3/89.8 | 97.2/93.3 |
> | Entropy-Min PI | 89.4/77.2 | 93.1/83.1 | 95.0/87.9 | 88.7/60.4 | 92.7/70.4 | 94.9/77.3 | 96.7/82.5 | 97.2/90.5 | 97.7/94.1 | 94.4/83.5 | 96.1/89.6 | 97.2/93.0 |
> | Entropy-Min MI+PI | 89.7/77.7 | 93.3/83.0 | 95.1/87.4 | 89.6/61.2 | 93.3/70.4 | 95.1/76.8 | 96.6/82.1 | 97.1/90.0 | 97.6/93.7 | 94.5/83.8 | 96.1/89.7 | 97.2/93.0 |
> | Cross-Entropy PI | 89.2/77.4 | 93.1/83.1 | 95.1/88.2 | 88.3/60.3 | 92.6/70.3 | 94.8/76.7 | 96.6/82.6 | 97.1/90.8 | 97.6/94.1 | 94.4/83.5 | 96.2/89.6 | 97.1/93.0 |
> | Cross-Entropy MI+PI | 89.6/77.7 | 93.3/83.3 | 95.0/87.4 | 89.5/61.1 | 93.1/70.6 | 95.1/76.8 | 96.7/83.7 | 97.1/90.7 | 97.7/93.9 | 94.5/83.8 | 96.2/89.9 | 97.1/93.1 |
> | IA+SA | 90.5/77.8 | 94.3/84.5 | 96.1/90.3 | 89.4/61.3 | 93.3/70.5 | 95.3/77.6 | 96.8/83.4 | 97.4/91.5 | 98.0/94.8 | 94.7/84.0 | 96.3/90.2 | 97.4/93.9 |
> | OAIMS (MI+PI) | 93.6/82.2 | 96.4/90.0 | 97.5/92.7 | 94.7/73.0 | 96.5/81.0 | 97.5/86.2 | 97.3/89.3 | 97.8/94.5 | 98.4/96.5 | 95.1/85.8 | 96.7/91.3 | 97.9/93.9 |
>
> **(B) Performance on Brain MRI (Dice %)**
>
> | **Method**| BRATS|  |  | |  |  |  |  |  |
> | :--- | :--- | :--- | :--- | :--- | :--- | :--- | :--- | :--- | :--- |
> | |T1 |||T1c|||T2|||
> | No. Clicks | 1 | 5 | 10 | 1 | 5 | 10 | 1 | 5 | 10 |
> | ICNN\* | 20.7 | 46.8 | 62.5 | 45.4 | 63.8 | 75.4 | 72.1 | 81.4 | 85.7 |
> | Entropy-Min PI | 20.1 | 48.2 | 65.6 | 45.4 | 65.3 | 76.3 | 72.6 | 82.0 | 86.5 |
> | Entropy-Min MI+PI | 13.8 | 38.1 | 57.1 | 42.3 | 57.8 | 66.8 | 70.0 | 81.2 | 86.8 |
> | Cross-Entropy PI | 20.0 | 49.7 | 66.0 | 46.6 | 65.6 | 76.7 | 72.5 | 82.1 | 86.6 |
> | Cross-Entropy MI+PI | 14.0 | 42.4 | 61.3 | 42.5 | 59.6 | 68.9 | 72.0 | 82.6 | 87.3 |
> | IA+SA | 28.6 | 57.0 | 70.6 | 48.3 | 67.1 | 78.2 | 76.5 | 84.1 | 87.9 |
> | OAIMS(MI+PI) | 71.2 | 83.4 | 88.0 | 70.4 | 82.9 | 87.5 | 84.9 | 90.6 | 93.0 |

---

> ### Author Response · Authors · 2025-11-23
>
> **\[Q1\]: “The role and necessity of multiple correction points are unclear, especially if Stage 1 user clicks are already assumed to be pseudo ground truth.”**
>
> If we understand this correctly, the reviewer questions why Stage 2 (Fine-tuning with Multiple Correction Points) is necessary given that Stage 1 (Fine-tuning with 1 Localization Click as input, and pseudo-ground truth as target) is already performed, since both stages have the same pseudo-ground-truth as the optimization target. We have justified this design through both empirical evidence and theoretical explanation.
>
> Empirically, we analyzed the contribution of Stage 2 in our ablation study (Table 5 in main paper). The last row of Table 5 represents using only Stage 1 ($S1\_{PI}$), while the fourth row represents the full Post-Interaction method, combining Stage 1 with Stage 2 (utilizing both $DFL\_{PI}$ and $CCGL\_{PI}$). The results demonstrate **a clear performance gap** (average 4.1%) , confirming that **Stage 2 provides crucial performance** **gains** that Stage 1 cannot achieve in isolation. Furthermore, the results show that combining both losses (Dice-Focal and CCG) in Stage 2 is important for consistent improvement across all datasets.
>
> Theoretically, Stage 1 and Stage 2 serve different purposes and use different input clicks. In Stage 1, the input is a single localization click, and the primary goal is to adapt the model such that it can make high-quality initial segmentation, with a single localisation click, so that as few corrections are needed later. In Stage 2, the input consists of multiple artificial correction clicks generated in erroneous regions. For interactive segmentation, it is insufficient to merely improve static segmentation accuracy; the model must also maintain its reactivity \---the ability to effectively leverage user clicks to correct specific regions (i.e., when a new input click is provided, the model must change the surrounding region based on this guidance). Under distribution shifts and long time online adaptation, this responsiveness must be learned and maintained.
> Stage 2 specifically targets this by retraining the model to react correctly to correction prompts (forcing the model to change the surrounding region to the target class when a click is present). This distinct goal explains why the Click-Centered Gaussian (CCG) loss is critical in Stage 2\. As a result, the whole Stage 2 not only helps the model learn from the Pseudo- Ground Truth (pseudo-GT) but, more importantly, reinforces the model's ability to react to points. It is important to emphasize that the Pseudo-GT for both stages remains the same: the final high-quality mask ($P\_{\text{final}}$) obtained from the user interaction process. This is clarified in the revised text in L211 and L219.
>
> We hope this clarifies this for the reviewer, and will now agree with other reviewers who have explicitly acknowledged that the method design is clear and well motivated (Rev. MLug “Stage two… with multiple correction points is well explained”, Rev. n4wo “conceptually clear and technically well-structured.” and “solid”). **To improve the manuscript, we have revised various points of the explanations in Sec. 2.4, as well as explanations in Fig.1 caption**, which should improve clarity even further for the readers.

---

> ### Author Response · Authors · 2025-11-23
>
> **\[Q2\]: “The introduction of artificial clicks risks injecting label noise during re-training under the stated loss functions, and this potential degradation should be analyzed and controlled. “**
>
> The reviewer inquires whether potential noise in the learning process due to sub-optimal Pseudo-Ground Truth (Pseudo-GT) and whether generating artificial clicks from this noisy label might potentially lead to performance degradation.
>
> The original paper included an assessment of the model's performance when only 3 clicks are given per image, in Table 4\. In this setting, without adaptation, the ICNN\*, which is the backbone of our method but without the online learning mechanism, achieves rather low the resulting Pseudo-GT contains significant errors (thus label noise) as indicated by the low Dice (44.6 \- 58.9%) when using only 3 clicks, which means they contain significant errors. Thus when using online learning in OAIMS, it is called to adapt using pseudo-GT of initially similarly low quality, hence containing significant label noise. Our method OAIMS manages to recover better performance, even with just 3 clicks, and consistently outperforms the non-adaptive baseline and previous adaptive method TSCA. This is a first indication of the models ability to handle artificial clicks generated from imperfect, noisy pseudo-labels.
>
> To further explore robustness in the case when pseudo-labels contain many errors, **we added new experiments in Section 3.3 with extreme domain shift.** We trained the model *only* on BraTS Flair and tested it on Atlas T1. This represents a very large domain shift where initial predictions without online adaptation are extremely poor (starting \<10% Dice with 1 click). The results of this experiment are shown in the below table.
> First, we examine performance when only 3 clicks are performed per image. Just with 1 localisation click and no online adaptation, the **base interactive model** (ICNN\*) achieves 9.3% Dice, very poor segmentation. After 3 clicks, it still performs very poorly (12.8% Dice), although it does improve slightly with each click. When we add our method’s online learning functionality to this base model, and use those very bad segmentations as pseudo-ground truth, the method still shows significant improvement. For example, with 3 clicks, online learning leads up to 57.9% Dice. This demonstrates that the very poor initial pseudo-GT (of 12.8% with 3 clicks) does not destroy the online learning process.
> We repeat the experiment but allow 10 clicks per image. This simulates a more realistic scenario than the above, when the user uses more corrections to improve an initially extremely bad segmentation. Our method of online learning recovers a high 80% Dice score, which is remarkable  taking into account that it has to learn from pseudo-GT of very bad initial quality. We note that these improvements are much greater than the strongest prior method, TSCA.
> In all these experiments, the model successfully adapted and improved, performing significantly better than previous methods, which shows that severe errors (noise) in labels does not strongly influence the performance .
>
> | Method | 1 click | 3 Clicks | 10 Clicks |
> | :---- | :---- | :---- | :---- |
> | **Base interactive model (No Adapt)** | 9.3 | 12.8 | 24.3 |
> | **TSCA** | 9.3 | 40.7 | 65.6 |
> | **OAIMS (Ours)** | **25.6** | **57.9** | **80.0** |
>
> **We improved the manuscript by adding the above experiments in Section 3.3, Table 10 (top),** along with further experiments with bad initial predictions and incorrect clicks (Table 10 bottom, 11), which additionally demonstrate the robustness of our method.

---

> ### Author Response · Authors · 2025-11-23
>
> **\[Q3\]: “The click centered Gaussian loss lacks sufficient technical detail, and the paper does not provide empirical insights or ablations to explain its behavior and sensitivity.”**
>
> With all due respect, this may be a misunderstanding of the reviewer, because some information and experiments in the paper may have been missed during the review. Allow us to clarify:
> Regarding technical details, the exact equation for the Click-Centered Gaussian (CCG) loss, including the Gaussian kernel definition and the class-indicator function, is already provided in **Section 2.3 of the original paper.**
> Regarding empirical insights and ablation study on CCG, the paper includes a study **in Table 5** which demonstrates the contribution of the CCG loss  in Mid-Interaction (MI) and Post-Interaction (PI) separately ($CCGL_{MI}$, $CCGL_{PI}$ columns). E.g. comparing rows 2-3 and 4-5 demonstrates inclusion of CCG in MI or PIoffers clear increases in various settings. Furthermore, the original paper **evaluated specific design aspects of CCG in Table 7**, which compares the Gaussian kernel against a uniform square kernel of the same size, and the class-specific mechanism against applying the loss to all surrounding pixels. **We added a sentence in Sec 3.2 to further clarify this ablation study.**
>
> To further enhance the study of CCG, we added a new ablation study on the σ of the CCG Loss in **Appendix A.8** of the revised manuscript (noting that σ close to 0 is essentially a loss focusing at a single-pixel).
>
> Collectively, these ablation studies demonstrate the effectiveness of our design.
> We hope that the above clarifications, pointers, and addition, resolve this point for the reviewer.
>
> **\[Q4\]: “The use of pretraining and its fidelity to real deployment are under specified, including whether the source model was trained with extensive click supervision and which loss functions were used.”**
>
> Respectfully, this may be a misunderstanding, as these details are already in the original paper. The loss function used is shown in Section 2.2: “we train the base model using simulated clicks and a compound loss: Dice–Focal (Eq. 4\) and CCG Loss (Eq. 3), which strengthens the model’s response to user clicks”. Additionally, we provide details regarding click simulation for training and the number of clicks used in Appendix A.2, which is referenced in Section 2.2 :“See Appendix A.2 for details regarding click simulation”.
>
> **\[Q5\]: “A concise algorithmic description for both source training and test time inference with precise mathematical notation would significantly improve reproducibility.”**
>
> We have added an **algorithmic description** (presented as pseudo-code) in **Appendix A.7** of the revised paper. This section details both the source training procedure and the test-time inference/adaptation loop.
> To further improve the main paper and make it even easier for the reader to understand the method, we also **revised the explanations in Fig 1 caption, and Section 2.4**, including introducing further math notation for clarity and disambiguity. While reviewers have provided positive comments about the clarity of the paper, we believe this makes reading easier for the readers, which is always beneficial. **For full reproducibility, as stated in the abstract, we will publish the code** of the method after publication.

---

> ### Author Response · Authors · 2025-11-23
>
> **\[Q6\]: “Extending results to adaptation oriented segmentation benchmarks would strengthen claims of generality and applicability.”**
>
> Respectfully, the experiments we have designed are adaptation-oriented segmentation benchmarks. We **include 9 databases** which enable studying multiple types of domain shifts. The experiments include **smaller domain shifts** between the source and test domains, leading to slight performance degradation when a model trained on one is tested on the other, **and very strong domain shifts** that lead to very large performance degradation, as can be seen in Figure 2 (indicated by the two horizontal lines in each subfigure). Besides being a challenging benchmark, evaluating the method using this medical data also allows us to assess the method in **clinically meaningful settings and tasks, of high practical importance**. Other benchmarks, e.g. on natural-image data, could have been chosen, but they have other advantages/disadvantages. Instead of spending resources on evaluating on even more databases, we instead spent this effort on conducting a deeper study and extensive ablation study, to provide rich insights about the method’s behaviour. Overall, reviewers have commented that the work includes **comprehensive evaluation, convincingly demonstrating the effectiveness** of the method (MLug: “Experiments cover a wide range of datasets …consistent cross-domain generalization.“; oQJi: “Experiments are comprehensive… significant performance gains”; KBrJ: “Comprehensive experiments… consistent improvements”), including acknowledged in your own review (in summary: “**Comprehensive experiments** across various medical imaging datasets demonstrate **consistent improvements**”). We hope this clarifies that the choice of data for evaluation is appropriate and the results are convincingly very promising.
>
> **\[Q7\]: “In Table 5 for BraTS, the inclusion of CCFL yields a large jump from 46.6 to 81.9, whereas similar gains are not observed elsewhere and are only marginal on BraTS T2, which warrants deeper analysis.”**
>
> Different components offer different benefits, which are more observable in different settings and types of data. This is why we evaluate on multiple settings using multiple databases. BRATS T2 shows improvements of 3-4% Dice, which is a very meaningful difference for this segmentation task, by addition of all modules in comparison to baseline, though no specific module makes a massive difference. Since the specific case you mention in BRATS T1 is the largest difference in the Table it is meaningful to comment on it further: BraTS T1 images are characterized by low contrast, which results in ambiguous tumor boundaries, making them significantly harder to segment. Consequently, when a click is provided, the changes can easily extend beyond the true boundary. If *only* the Dice-Focal loss is used, the model can be hindered by this incorrect information (as evidenced by the drop from 78.9% to 48.9% between row 4 and 3 in Table 5). However, using the Click-Centered Gaussian (CCG) loss directs the update towards pixels closer to the user click, which are more likely to be correct and thus more reliable, mitigating such issue and leading to high performance (row 1 and 2). **We add this explanation to the ablation study in the Section 3.2 of the paper.**
>
> **\[Q8\]: Unsupervised losses:** answered with \[W2\]
>
>
> **\[Q9\]: “Clarification is needed on which network layers are updated during adaptation “**
>
> We clarify that all model parameters are updated during our online adaptation. We do not set any frozen layers, because that setting restricts the implementation of our OAIMS method to specific architectures. Even with fine-tuning on all parameters, our method performs better without showing observable overfitting, thanks to the robust design of our framework. **We added this clarification in section 2.4  and section 3.2 of the revised paper.**
>
> ---
>
> Thank you again for your effort in reviewing our work. We hope we have addressed all the points you raised and that this helps strengthen your support for the publication of this paper. If you have any questions, please let us know.

---

> ### Comment · Reviewer_KBrJ · 2025-11-23
>
> Thanks for the efforts and the detailed answers. While I am going through them, I have a few quick follow-up questions:
>
> - Can the authors provide more experimental details for the usage of the new Appendix A10 unsupervised loss and pseudo-label-based adaptation? Specifically, if losses from established works were used, could the papers be listed
> - [Q6]: I would like to clarify that I was referring to common natural vision-based segmentation benchmarks that are designed for segmentation and adaptation. For instance: GTA5 to Cityscapes, Synthia to Cityscapes.
> - [Q9]: Thanks for clarifying that all layers were used. Were any additional experiments carried out to analyze the layer-wise effect on adaptation?
> - [Q2]: For the experiments with extreme domain shift, could the authors cite works which also align with the finding of your experiments: 'which shows that severe errors (noise) in labels do not strongly influence the performance.'

---

> > ### Author Response · Authors · 2025-11-28
> > **Response to follow-up (part 1)**
> >
> > We hope you have found the response easy to read and the manuscript updates valuable, as a lot of effort was paid in addressing all comments appropriately. Thank you for engaging, and your followup questions:
> >
> > **[Q8] followup:** “Can the authors provide more experimental details for the usage of the new Appendix A10 unsupervised loss and pseudo-label-based adaptation? Specifically, if losses from established works were used, could the papers be listed”
> >
> > Absolutely. We now further updated Appendix A.10 with more details. Apologies, but because Entropy Minimization and CE with pseudolabels are fundamental, we assumed the readers would be familiar with them. But you are right, this may not be true for many readers. Therefore we extended the description and added references.
> >
> > Specifically, following your suggestion, we experimented with the **original Entropy Minimization [1] and pseudo-label training [2], but integrated them into our framework’s MI and PI for meaningful comparison**. Our team has a decade of experience with these methods in semi-supervsied learning and domain adaptation and we are confident in their implementation. We refer the reviewer to the revised Appendix A.10 for the detailed explanations. In short, we introduced references, math notation, equations for each method, and explanations how they are integrated into MI and PI of our framework. We believe this improved readability of A.10. Thank you.
> >
> > **[Q6] followup:** “I would like to clarify that I was referring to common natural vision-based segmentation benchmarks that are designed for segmentation and adaptation. For instance: GTA5 to Cityscapes, Synthia to Cityscapes.”
> > **[Q6] original request, for easy reference:** “Extending results to adaptation‑oriented segmentation benchmarks would strengthen claims of generality and applicability.”
> >
> > Thank you for clarifying, we understand this and the original question.
> > With respect, we do not think this suggestion is a good fit for our work. Our original answer addressed this but allow us to further elaborate in response to your follow-up:
> >
> > - The suggested benchmarks, GTA5 (video game), SYNTHIA (synthetic cities) and Cityscapes, or similar vision benchmarks, are **out of scope** of our work. From title, abstract, intro, to conclusion, the scope of the project is defined as tackling a real-world, and very impactful challenge in medical imaging. **Distribution shifts in medical imaging** are a main obstacle for deploying AI in healthcare. AI can have **tremendous impact by accelerating the work of radiologist**, who are overwhelmed due to staff shortages. Also, healthcare requires **\*interactive\* AI methods** that clinicians can use, rather than automated methods that replace them (for safety). Therefore, this application is extremely fitting for studying methods that are interactive and tackle domain-shift. Hence, natural image benchmarks do not fit well the scope of the study.
> > Note that **reviewers acknowledged explicitly the importance of the medical application** (E.g. Reviewer n4wo: “The paper tackles a **practically important problem**, handling distribution shifts in **medical image**”).
> >
> > - The **benchmarks we employed are commonly used to study domain-shift \*and\* interactive segmentation method**s. Studied prior methods TSCA (Fundus, MRI, CT), IA+SA (Fundus), and ICNN\* (CT) **have all used medical imaging** for evaluation. The benchmarks referenced by the reviewer, while they are common for “standard” domain-adaptation, they are not  commonly used for \*adaptive interactive segmentation\* (we found no prior work in this space using them). Employing them could pose unexpected gargantuan challenges in even configuring existing baselines to work appropriately (IA+SA, TSCA, ICNN\* not made for them), without adding value to the paper proportional to the required effort (see next point).
> >
> > - We already evaluated on **9 databases of Fundus imaging and Brain MRI, across numerous types of distribution shifts**. Reviewers have acknowledged **performance gains over existing methods are clear**, including your review (“**Comprehensive experiments**…demonstrate **consistent improvements** over existing methods“ and “evaluation spans **multiple levels of distribution shift**”). **Other reviewers also agree performance is convincing**: Rev MLug: “convincingly show that their method works better”, Rev n4wo: “consistent cross-domain generalization”. Rev oQJi: “significant performance gains over existing methods” ).
> >
> > While we agree that more evaluation is always welcome, we believe the evidence of performance is sufficient. We think the addition of 2-3 further databases (unrelated to medical scope) will not add too much value. Instead we preferred using our resources and paper space for rich analysis of method properties, such as ablations, sensitivity studies, etc (reviewers acknowledge they are plenty, now further extended in revision).

---

> > ### Author Response · Authors · 2025-11-28
> > **Response to follow-up (part 2)**
> >
> > **[Q9] follow-up:** "Thanks for clarifying that all layers were used. Were any additional experiments carried out to analyze the layer-wise effect on adaptation?”
> >
> > We have not performed such experiments, and **we do not see a reason why such analysis is particularly relevant to this work** to be prioritized. We note that the reviewer has not asked for such experiments either, just for a clarification in the original review (“Clarification is needed…”).
> > The reason why we do not think such analysis is particularly relevant is because fine-tuning (updating) specific layers of a model, while keeping other layers frozen, is usually motivated when a) there is need to accelerate fine-tuning (e.g. fewer and faster to update few parameters instead of whole model, PEFT etc); b) in cases when overfitting does not allow fine-tuning all parameters to perform well (limit parameter updates to few parameters so that they cannot overfit). However, our method is already fast (Sec 3.2, page 8: GPU: 0.05s for MI; 0.09s for PI; CPU: 0.25s for MI; 0.41s for PI). Also, since our performance clearly outperforms SOTA (acknowledged by reviewers, e.g. your summary: “consistent improvements over existing methods”, Rev MLug: “convincingly show that their method works better”, Rev n4wo: “consistent cross-domain generalization.” and “solid”. Rev oQJi: “significant performance gains over existing methods” ), there is no clear motivation why one would pursue layer-specific fine-tuning to further improve performance, and prioritize it over other directions/analysis. Given the method is clearly promising, we believe this warrants its publication so that the community has the chance to learn about it and then seek to further investigate directions that may push its performance further (incl. Layer-wise updates, which may or may not be a promising hypothesis).
> >
> > **[Q2] follow-up:** "For the experiments with extreme domain shift, could the authors **cite works** which also align with the finding of your experiments: 'which shows that severe errors (noise) in labels do not strongly influence the performance.”
> >
> > With respect, we do not think that the finding reported by this phrase warrants we should seek to cite prior work. The statement simply states our methods performs well in this setting. That’s all. Therefore we do not see how other works should be cited for an experiment that demonstrates our method’s performance.
> >
> > We wonder if the reviewer may have misunderstood this statement made in the rebuttal as if it is a generic statement about a family of methods (e.g. that \*all\* interactive online adaptation methods are robust to label noise?), and therefore reviewer wonders if other works have shown such a generic phenomenon? This is not what this phrase in the rebuttal response conveys, it conveys something very simple:
> >
> > The quoted phrase is from the rebuttal response (“the model successfully adapted and improved, performing significantly better than previous methods, which shows that severe errors (noise) in labels does not strongly influence the performance.”) and **it simply conveys that the experiments show that \*our\* method’s performance is good in this setting**, better than the compared methods, and that our method does not drastically suffer from label noise (“does not \*strongly\* influence the performance”), as it reaches high performance 80% under extreme conditions. Therefore we do not think citation to previous work for such a statement is appropriate.
> >
> > We also clarify that **the phrase that the reviewer quotes is used only within the rebuttal response, not in the manuscript.** In related text in Sec 3.3 of the manuscript, no ambiguous statements are made either (we checked).
> >
> > In case the reviewer is surprised about the high performance of our method and seeks reassurance, we commit that code for reproducing the method will be published after publication that reproduces these performances (as stated in the abstract). The authors take scientific integrity extremely seriously and reviewing is expected to be done in good faith that this is true (non-integrity can destroy careers, so, yes, we do take it extremely seriously).
> >
> > [1] Grandvalet and Bengio, Semi-supervised Learning by Entropy Minimization, NIPS 2004
> >
> > [2] Lee, Pseudo-label: the simple and efficient semi-supervised learning method for deep neural networks, Workshop on challenges in representation learning, ICML 2013\.

---

### Author Response · Authors · 2025-11-21
**Reponse to reviewers to be submitted soon**

Dear reviewers,

Thank you very much for your work in reviewing our paper and for the feedback.

This message is a heads-up, in case this helps plan your own workload. This is to let you know that we are planning to submit our response to all comments soon, hopefully tomorrow. We were planning for an earlier submission of the response but the senior author (me) recently got a baby and all workload is behind, including revising and confirming the response before its publication. The response will be with you soon.

Thank you once again for your time and effort.

---

### Meta-Review · Area_Chair_iTnQ · 2025-12-23

**Summary:**

The paper proposes an online adaptation framework for interactive medical image segmentation that leverages user clicks not only to refine predictions but also to update model parameters to better handle distribution shifts. It treats post-interaction refined outputs as pseudo-labels and performs both post-interaction and mid-interaction updates using a click-centered Gaussian loss to strengthen click responsiveness. Experiments on fundus and brain-MRI datasets show consistent improvements under unseen modalities and pathologies compared with existing methods.

Reviewers requested deeper technical discussion, additional baseline comparisons, stronger justification of key design choices, evaluations with human annotators, extensions to transformer-based architectures, and clearer implementation details. The authors’ rebuttal addresses most of these points comprehensively through added explanations and experiments. Overall, the revisions strengthen the technical and empirical case, supporting acceptance.

**Reviewer Concerns:**

Most of the reviewers’ concerns have been adequately addressed in the authors’ rebuttal.

**Reviewer Scores:**

There is only one negative review, and its score may increase since the rebuttal addresses most of the concerns raised by that reviewer.

---

### Decision · Program_Chairs · 2026-01-26

Accept (Poster)